# TimeSpec4LULC: A Global Multispectral Time Series Database for Training LULC Mapping Models with Machine Learning

Rohaifa Khaldi[1,5], Domingo Alcaraz-Segura[1,3], Emilio Guirado[4], Yassir Benhammou[5], Abdellatif El Afia[2], Francisco Herrera[5], and Siham Tabik[5]

[1]Dept. of Botany, Faculty of Science, University of Granada, 18071 Granada, Spain
[2]ENSIAS, Mohammed V University, Rabat, 10170, Morocco
[3]iEcolab, Inter-University Institute for Earth System Research, University of Granada, 18006 Granada, Spain
[4]Multidisciplinary Institute for Environment Studies "Ramón Margalef", University of Alicante, 03690, Spain
[5]Dept. of Computer Science and Artificial Intelligence, Andalusian Research Institute in Data Science and Computational Intelligence, DaSCI, University of Granada, 18071, Granada, Spain

**Correspondence:** Rohaifa Khaldi (rohaifa@ugr.es); Domingo Alcaraz-Segura (dalcaraz@ugr.es)

**Abstract.** Land Use and Land Cover (LULC) mapping are of paramount importance to monitor and understand the structure and dynamics of the Earth system. One of the most promising ways to create accurate global LULC maps is by building good quality state-of-the-art machine learning models. Building such models requires large and global datasets of annotated time series of satellite images, which are not available yet. This paper presents TimeSpec4LULC (Khaldi et al., 2022), a smart open-source global dataset of multi-spectral time series for 29 LULC classes ready to train machine learning models. TimeSpec4LULC was built based on the seven spectral bands of the MODIS sensors at 500 m resolution, from 2000 to 2021, and was annotated using spatial-temporal agreement across the 15 global LULC products available in Google Earth Engine (GEE). The 22-year monthly time series of the seven bands were created globally by: (1) applying different spatial-temporal quality assessment filters on MODIS Terra and Aqua satellites, (2) aggregating their original 8-day temporal granularity into monthly composites, (3) merging Terra+Aqua data into a combined time series, and (4) extracting, at the pixel level, 6,076,531 time series of size 262 for the seven bands along with a set of metadata: geographic coordinates, country and departmental divisions, spatial-temporal consistency across LULC products, temporal data availability, and the global human modification index. A balanced subset of the original dataset was also provided by selecting 1000 evenly distributed samples from each class such that they are representative of the entire globe. To assess the annotation quality of the dataset, a sample of pixels, evenly distributed around the world from each LULC class, was selected and validated by experts using very high resolution images from both Google Earth and Bing Maps imagery. This smartly, pre-processed, and annotated dataset is targeted towards scientific users interested in developing various machine learning models, including deep learning networks, to perform global LULC mapping.

# 1 Introduction

Broadly, land cover (LC) refers to the different vegetation types (usually following biotype, plant functional type, or physiognomy schemes, such as forests, shrublands, or grasslands) or other biophysical classes (such as water bodies, snow, or bare soil) that cover the Earth's surface (Moser, 1996). Land cover is an essential variable that provides powerful insights for the assessment and modelling of terrestrial ecosystem processes, biogeochemical cycles, biodiversity, climate, and water resources, etc. (Luoto et al., 2007; Menke et al., 2009; Polykretis et al., 2020). Whereas, land use (LU) incorporates many types of modifications that an increasing human population, more than 9 billion expected by 2050, causes to the LC (such as urban areas and croplands). Accurate LULC information, including distribution, dynamics and changes, is of paramount importance for understanding and modelling the natural and human-modified behavior of the Earth's system (Tuanmu and Jetz, 2014; Verburg et al., 2009).

LULCs are subjected to anomalies, trends and changes both from anthropogenic and natural origins (Polykretis et al., 2020). LULC change is usually interpreted as the conversion from one LULC category to another and/or the modification of land management within LULC (Meyer et al., 1994). LULC is an essential climate and biodiversity variable (Bojinski et al., 2014; Pettorelli et al., 2016) to model and assess the status and trends of social-ecological systems from the local to the global scale in the pursuit of a safe operating space for humanity (Steffen et al., 2015). For example, characterizing such LULC changes is critical for the climate through two mechanisms: biophysical (BPH) and biogeochemical (BGC) feedbacks (Duveiller et al., 2020). For instance, the conversion from forests to croplands (i.e., deforestation) generates a fast increase land surface temperature (i.e. biogeophisical effect) and also releases part of the carbon stored in the forest into the atmosphere (i.e. biogeochemical component). Both mechanisms contribute to local and global warming, respectively (Oki et al., 2013). Other examples of LULC conversion are urban sprawl, agriculture expansion or abandonment, which also affect the biodiversity, soil and water quality, food security, and human health among many others (Lambin and Geist, 2008; Feddema et al., 2005). For these reasons, continuous and accurate LULC and LULC change mapping is essential in policy and research to monitor ecological and environmental change at different temporal and spatial scales (Polykretis et al., 2020; García-Mora et al., 2012), and as a decision support system to ensure an effective and sustainable planning and management of natural resources (Kong et al., 2016; Congalton et al., 2014; Grekousis et al., 2015).

Satellite remote sensing in combination with geographic information systems (GIS) have provided convenient, inexpensive, and continuous spatial-temporal information for mapping LULCs and detecting changes on the Earth's surface from regional to global scales (Kong et al., 2016; Kerr and Ostrovsky, 2003; Pfeifer et al., 2012) thanks to their strong ability to cover, timely and repeatedly, wide and inaccessible areas, and to get high spatial and temporal resolution data (Alexakis et al., 2014; Yirsaw et al., 2017; Patel et al., 2019).

Deep Learning (DL), a sub-field of machine learning essentially based on deep artificial neural networks (Zhang et al., 2018c), has shown impressive performance in computer vision and promising ones in remote sensing during the last decades. Currently, two specific types of DL models, i.e., CNNs (Convolutional Neural Networks) and RNNs (Recurrent Neural Networks), constitute the state-of-art in respectively extracting spatial and temporal/sequential patterns from data-records. Indeed,

DL models are showing great performance in LULC tasks such as scene classification (Zhang et al., 2018a), object detection

(Zhao et al., 2015; Guirado et al., 2021) and segmentation (Zhao and Du, 2016; Guirado et al., 2017; Safonova et al., 2021) in RGB and multi-spectral satellite and aerial images. However, such good performance is only possible when DL models are trained on smart data. The concept of smart data involves all pre-processing methods that improve value and veracity of data and of associated expert annotations (Luengo et al., 2020), resulting in high quality and accurately annotated datasets. In general, remote sensing datasets contain noise, missing values, and high variability and complexity across space, time, and

spectral bands. Applying pre-processing methods, such as gap filling and noise reduction to data, and consensus across multiple sources to annotations, contribute to creating smart remote sensing datasets.

Currently, there only exist few multi-spectral datasets annotated for training DL models to map LULC and monitor their change (Table 1). However, most of these datasets provide very short time series of data, very few LULC classes, and do not have a global coverage. As far as we know, there is no dataset designed for DL models that allows global scale analysis of

65 many LULC classes using long time series data.

This paper presents, TimeSpec4LULC, a new open-source, smart, and global dataset of multi-spectral time series targeted towards the development and evaluation of DL models to globally map LULCs. TimeSpec4LULC was built using GEE (Gorelick et al., 2017) by combining the seven 500m spectral bands of MODIS Aqua and MODIS Terra satellite sensors at a monthly time step from 2000 to 2021. It contains millions of pixels that were annotated based on a spatial-temporal consensus across up

to 15 global LULC products (Table 2) for 29 broad and globally harmonized LULC classes. In addition, it provides a metadata at pixel level: geographic coordinates, country and departmental divisions, spatial-temporal consistency across LULC products, statistics on temporal data availability, and the global human modification index. The annotation quality was further assessed by experts using Google Satellite and Bing Maps very high resolution images using 100 samples per class evenly distributed around the world.

**Table 1.** A list of existing datasets of times series of satellite images, including the proposed TimeSpec4LULC dataset, for training machine learning models.

| Dataset | Source | # images × (pixels) | Spatial Resolution (m) | Temporal Resolution | No. Bands | No. Classes | Extent | Intra/Inter time series | labeled for |
|---|---|---|---|---|---|---|---|---|---|
| CaneSat (Virnodkar et al., 2020) | Sentinel-2 | 1627× (10×10) | 10 | Monthly | 6 | 2 | India | [2018, 2019] | Sugarcane classification |
| SpaceNet-7 (Van Etten et al., 2021) | Dove Satellite Constellation Planet Labs' | 24× (1024×1024) | 4 | Monthly | 8 | 2 | 100 cities | [2017,2020] | Buildings tracking |
| Time series spectral dataset for croplands in France (Hubert-Moy et al., 2019) | MODIS and LPIS | 21,129 pixels | 250 | 8-day intervals | 4 | 19 | France | [2006, 2017] | Crop type mapping and monitoring |
| TiSeLaC (TiS) | Landsat | 8 × (2866× 2633) | 30 | Annually | 10 | 9 | The Reunion Island | 2014 | classification |
| BreizhCrops Rußwurm et al. (2019) | Sentinel-2 | 610,000 pixels | 60 | - | 10 | 9 | Brittany dept. France | [01/01/2017, 31/12/2017] | Crop type mapping |
| **TimeSpec4LULC (Ours)** | MODIS | 11,856,992 pixels | 500 | Monthly | 7 | 29 | Global | [03/2000, 12/2021] | LULC mapping |

## 2 Methods

To build TimeSpec4LULC, we first determined the spatial-temporal agreement across 15 heterogeneous global LULC products (listed in Table 2) for 29 broad and globally harmonized LULC classes. Then, for each class, we extracted a 22-year monthly time series for the seven 500-meter spectral bands of MODIS Terra and Aqua combined. We carried out this process in GEE since it provides access to freely available satellite imagery under a unified programming, processing and visualization environment.

**Table 2.** Description of the GEE global LULC products used in this study.

| Product ID(s) | Products | Version | Provider | Sensor | Satellite or Spaceborn | Spatial resolution | Acquisition time | Data type | Link | Ref. |
|---|---|---|---|---|---|---|---|---|---|---|
| P1:P5 | MCD12Q1 (LC Type 1 to 5) | v6 | NASA LP DAAC at the USGS EROS Center | MODIS | Aqua-Terra | 500 m | 2001-2019 | Image collection | (MCD) | Sulla-Menashe and Friedl (2019) |
| P6 | CGLS-LULC100 | v3.0.1 | Copernicus Global Land Service (CGLS) | | PROBA-V | 100 m | 2015-2019 | Image collection | (CGL) | Buchhorn et al. (2020) |
| P7 | GFCC | v3 | NASA LP DAAC at the USGS EROS Center | Multi-sensor | Multi-satellite | 30 m | 2000, 2005, 2010, 2015 | Image collection | (GFC, a) | Sexton et al. (2013) |
| P8 | GLOBCOVER | v2 | ESA and by the Catholic University of Louvain | MERIS | ENVISAT | 300 m | 2009 | Single image | (Glo) | Arino et al. (2008) |
| P9 | GFSAD | v0.1 | Global Food Security support Analysis Data at 30m Project (GFSAD30) | Multi-sensor | Multi-satellite | 1000 m | 2010 | Single image | (GFS) | Teluguntla et al. (2015) |
| P10 | PALSAR2 | vfnf | JAXA EORC | SAR | ALOS, ALOS2 | 25 m | 2007-2010 2015-2017 | Single image | (PAL) | Shimada et al. (2014) |
| P11 | HANSEN | v1.7 | Hansen, UMD, Google, USGS, NASA | OLI | Landsat 8 | 1" | 2000-2019 | Single image | (Han) | Hansen et al. (2021) |
| P12 | GFCH | v2005 | NASA, JPL | | Lidar | 30" | 2005 | Single image | (GFC, b) | Simard et al. (2011) |
| P13 | JRC Yearly Water Classification History | v1.2 | EC JRC, Google | Multi-sensor | Landsat (5, 7, 8) | 30 m | 1984-2019 | Image collection | (JRC, b) | Pekel et al. (2016) |
| P14 | JRC Global Surface Water Mapping Layers | v1.2 | EC JRC, Google | Multi-sensor | Landsat (5, 7, 8) | 30 m | 1984-2019 | Single image | (JRC, a) | Pekel et al. (2016) |
| P15 | Tsinghua FROM-GLC | v10 | Tsinghua University | Multi-sensor | Landsat | 30 m | 1985-2018 | Single image | (Tsi) | Gong et al. (2020) |

### 2.1 Finding spatial-temporal agreement across 15 global LULC products

Since the 1980s, multiple global LULC products (Table 2) have been derived from remotely sensed data, providing alternative characterizations of the Earth surface at varying extents of spatial and temporal resolutions (Townshend et al., 1991; Loveland et al., 2000; Bartholome and Belward, 2005). One of the most important limitations of global LULC products is the within-product variability of accuracy (across different years, regions, and LULC types), and the low agreement among products in many regions of the world (Tsendbazar et al., 2015b, 2016; Gao et al., 2020; Gong et al., 2013; Zimmer-Gembeck and Helfand, 2008). The accuracy of the global products at the local level is low compared to their accuracy at the global level and to the accuracy of local products at local level. Such lack of consensus can translate into huge implications for subsequent global

assessments of biodiversity status, carbon balance, or climate change (Estes et al., 2018; de la Cruz et al., 2017). In addition, accuracy at the local level can be too low, which impedes the use of global or regional LULC products in local studies (Hoskins et al., 2016; Tsendbazar et al., 2016), since it can lead to different conclusions due to the compelling amount of inconsistencies, uncertainties, and inaccuracies (Tsendbazar et al., 2015a; Estes et al., 2018). Multiple reasons lay behind these discrepancies among LULC products (Congalton et al., 2014; Grekousis et al., 2015; Gómez et al., 2016):

– Satellite sensors: the spatial, temporal and spectral resolutions of the source satellite images strongly determine the precision and accuracy of derived LULCs. Native pixel size can vary from dozens of meters to kilometers, which determines the precision. Revisiting frequency can vary from daily images to several weeks, which determines the possibility of removing cloud and atmospheric noise effects. In addition, the greater the number of spectral bands in a sensor, the greater the amount of complementary information that can help to differentiate among LULC classes.

– Processing techniques: the different algorithms for atmospheric correction, cloud filtering, image composition, viewing geometry corrections, etc. can also influence LULC accuracy.

– Acquisition year(s): some LULC products just refer to a particular year while others are regularly updated.

– Classification schemes: LULC legends can greatly differ in the number of classes and typology definitions. In general, LULC products tend to agree more in broader general categories than in finer specific ones.

– Classification algorithms: the approaches and rules used to identify each LULC have evolved from decision trees, to multivariate clustering, and machine learning, including now deep learning.

– Validation techniques of the final product. The amount and global distribution of ground truth samples differs across products and influences their reported accuracy.

Many efforts have been made to assess, compare, and harmonize the increasing plethora of global, regional and local LULC products, including their integration into synthetic products, which has shed light onto their strengths and weaknesses (Feng and Bai, 2019; Zhang et al., 2019; Gao et al., 2020; Liu et al., 2021). Still, the myriad of existing products with different specifications and accuracies have made their selection by the users problematic, and discouraging because it is frequently unknown whether a product meets the user needs for a particular area or LULC class (Tsendbazar et al., 2015b; Xu et al., 2020). In addition, many of these efforts are either limited to regional or national scale (e.g. Pérez-Hoyos et al. (2012); Gengler and Bogaert (2018)), coarse spatial resolution (e.g. Tuanmu and Jetz (2014); Jung et al. (2006)), or just one LULC type (e.g. Fritz et al. (2011)). The use of synergistic products takes advantage from the strengths of individual products while attenuating their respective weaknesses. However, they still face the challenge of taking into consideration the spatial-temporal consistency within pixels. In general, given a target maximum error of 5–15% either per class or for the overall accuracy, most of the current global land-cover maps still do not meet the accuracy demands of many applications (Liu et al., 2021).

To overcome all the aforementioned limitations, a spatial-temporal agreement across 15 global LULC products available in GEE was performed. To find the spatial-temporal consensus across global LULC products for different LULC classes,

we followed five steps: 1) selection of global LULC products, 2) standardization and harmonization of LULC legends, 3) combination of products across space and time, and 4) reprojection and selection of spatial agreement thresholds to get a final consistent mask across the 15 products for each one of the 29 LULC classes.

### 2.1.1 Selection of global LULC products

We used the 15 most updated global LULC products available in GEE (Table 2). These products widely differ in their source satellite data, spatial resolution, temporal coverage, class legend, and accuracy. Given such heterogeneity, we used the consensus across all of them in space and time as a source of reliability to support our annotation. That is, a given LULC class is assigned to a 500 m pixel only if it was consistent over time and space across all the 15 LULC products.

### 2.1.2 Standardization and harmonization of LULC legends

To standardize and harmonize the LULC legends across the 15 LULC products, we used expert knowledge (Vancutsem et al., 2013) to find a common nomenclature based on spatial, temporal, and thematic consensus between equivalent classes from different products. We always matched our resulting consensus class into the hierarchy of FAO's Land Cover Classification System (LCCS) (Di Gregorio, 2005); see correspondence across LULC products in Table 4, and the correspondence with FAO's LCCS in Table A1 of the appendix. Our final legend contained 29 classes at the finest detail (6 LU classes and 23 LC 135 classes) that were interoperable across all products (see the hierarchical structure of our legend in Fig. 1) and to FAO's LCCS (Table A1). Table 3 provides the IDs, full names, and short names of the 29 LULC classes. Table A2, in Appendix, provides the detailed definitions of each one of the 29 classes from the definitions given in the original products.

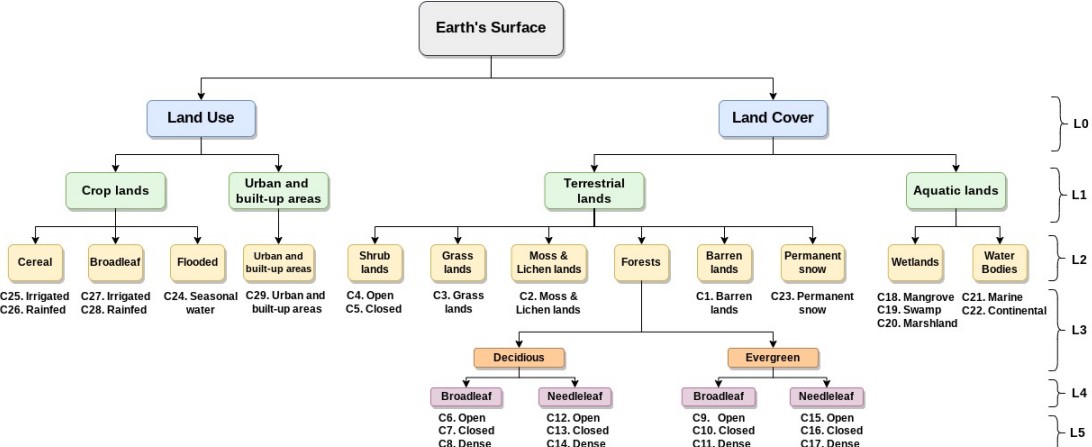

**Figure 1.** Hierarchical structure of the LULC classes contained in the TimeSpec4LULC dataset. C1 to C29: the 29 LULC classes. L0 to L5: the 5 LULC levels. L0 includes the 2 blue boxes. L1 includes the 4 green boxes. L2 includes the 12 yellow boxes. L3 includes all the classes of the 12 yellow boxes (from C1 to C5 and from C18 to C29) except the Forests class where it includes only the 2 orange boxes (Decidious and Evergreen). L4 includes the same classes but expands the Forests class into the 4 purple boxes: Decidious (Broadleaf and Needleleaf) and Evergreen (Broadleaf and Needleleaf). L5 includes all the 29 LULC classes (from C1 to C29).

**Table 3.** Description of the full name and short name of each LULC class in the TimeSpec4LULC dataset.

| Class Id | Class full name | Class short name |
|----------|-----------------|------------------|
| C1 | Barren Lands | BarrenLands |
| C2 | Moss and Lichen Lands | MossAndLichen |
| C3 | Grasslands | Grasslands |
| C4 | Open Shrublands | ShrublandOpen |
| C5 | Closed Shrublands | ShrublandClosed |
| C6 | Open Deciduous Broadleaf Forests | ForestsOpDeBr |
| C7 | Closed Deciduous Broadleaf Forests | ForestsClDeBr |
| C8 | Dense Deciduous Broadleaf Forests | ForestsDeDeBr |
| C9 | Open Deciduous Needleleaf Forests | ForestsOpDeNe |
| C10 | Closed Deciduous Needleleaf Forests | ForestsClDeNe |
| C11 | Dense Deciduous Needleleaf Forests | ForestsDeDeNe |
| C12 | Open Evergreen Broadleaf Forests | ForestsOpEvBr |
| C13 | Closed Evergreen Broadleaf Forests | ForestsClEvBr |
| C14 | Dense Evergreen Broadleaf Forests | ForestsDeEvBr |
| C15 | Open Evergreen Needleleaf Forests | ForestsOpEvNe |
| C16 | Closed Evergreen Needleleaf Forests | ForestsClEvNe |
| C17 | Dense Evergreen Needleleaf Forests | ForestsDeEvNe |
| C18 | Mangrove Wetlands | WetlandMangro |
| C19 | Swamp Wetlands | WetlandSwamps |
| C20 | Marshland Wetlands | WetlandMarshl |
| C21 | Marine Water Bodies | WaterBodyMari |
| C22 | Continental Water Bodies | WaterBodyCont |
| C23 | Permanent Snow | PermanentSnow |
| C24 | Croplands Flooded with Seasonal Water | CropSeasWater |
| C25 | Irrigated Cereal Croplands | CropCereaIrri |
| C26 | Rainfed Cereal Croplands | CropCereaRain |
| C27 | Irrigated Broadleaf Croplands | CropBroadIrri |
| C28 | Rainfed Broadleaf Croplands | CropBroadRain |
| C29 | Urban and Built-Up Areas | UrbanBlUpArea |

The LULC legend was structured into 6 hierarchical levels (L0 to L5). The six anthropogenic LU classes contained Urban and Built-Up Areas and five types of croplands. The 23 natural or semi-natural LC classes covered 5 aquatic systems (Marine water bodies, Continental water bodies, and 3 types of Wetlands) and 18 terrestrial systems (Permanent snow, Barren lands, Moss and Lichen lands, Grasslands, Closed Shrublands, Open Shrublands, and 12 types of Forests that differed in their canopy type, phenology and tree cover).

Some of the products provide discrete categorization of LULC classes in each pixel (P1-P5, P8-P10, P13, and P14), while other products provide continuous categorization represented by a class proportion in each pixel (P11, P12, and P15), or even both continuous and discrete categorizations of LULC (P6 and P7) (Table 4). To define the class of each pixel within these two different categorization mechanisms, we either specify a unique value (e.g., select the value 16 to access barren lands in P1) or use a range of values (e.g., Tree Canopy Cover less than 10 ($TCC < 10$) to access barren lands in P6).

### 2.1.3 Combining products across time and space

For each LULC class, we built a consensus image describing its global distribution by agreement over time and space across the LULC products. Based on their data type, the LULC products can be classified into two main categories: (1) products with single image referring to a particular year or period (P1 to P7), and (2) products with a collection of images over years (P8 to P12). Thus, the temporal agreement can only be applied for the second category of products. In (1) the single image based products, we obtained a binary mask where value 1 corresponds to the targeted LULC class. Whereas, in (2) the image collection based products, we first obtained a binary mask for each year, then we produced their combination over years to obtain one mask. Afterwards, we performed a spatial agreement over the 12 masks of the first 12 products (P1 to P12), then we used the masks of the two water bodies products (P13 and P14) and the mask of the impervious surface product (P15) to further refine the consensus.

Based on the temporal consistency, the LULC classes can be classified into: (1) Classes with high temporal stability, namely Urban and Built-Up Areas, Water Bodies, Permanent Snow, Open Shrublands, Barren Lands, and Grasslands. (2) Classes with low temporal stability characterized with plausible inter-annual changes, namely Moss and Lichen Lands, Forests, Closed Shrublands, Wetlands, and Croplands. This instability is due to several reasons, for example wetlands affected by droughts, or large areas of no-forest cover in one year preceded and followed by forest in the previous and following years, respectively. Our main objective is to collect from each class a representative number of pixels that satisfy the temporal stability constraint of a specific class type. Thus, the temporal agreement, over the masks of each image collection based product, was performed based on two different types of operators governed by Algorithm 1: (1) The AND operator, which represents a hard temporal stability constraint, ensures getting pixels with stable class type over time but more likely small number of pixels. (2) The MEAN operator, which represents a soft temporal stability constraint, provides a large number of pixels, however, with less stability pattern over time. A list of these operators mapping each LULC class is provided in Table 5.

Subsequently, the spatial combination of the 15 masks was performed following six rules according to the global abundance of each class. The main rule (Rule 1) is to apply the MEAN operator across products P1 to P12 and multiply the result by the two water masks of P13 and P14 to eliminate water pixels from land classes and land pixels from water classes, and by the

impervious surface mask of P15 to eliminate impervious pixels from all classes but urban. However, when the number of pixels for some LULC classes is small (less than 1000), the Rule 1 was relaxed differently generating 5 other different rules (Rule 2 to Rule 6). These five rules were applied to five LULC classes that had too few pixels with Rule 1: the Moss and Lichen Lands (Rule 2), Mangrove Wetlands (Rule 3), Swamp Wetlands (Rule 4), Marshland Wetlands (Rule 5), and Croplands Flooded with seasonal water (Rule 6). The usage of the spatial combination rules is described in Algorithm 2.

Finally, the spatial-temporal combination of the 15 LULC products resulted in a mask for each LULC class produced at the resolution of the finest product (i.e., 30 m), where each pixel had a consensus level value $p$ in $[0, 1]$. Hence, for each LULC mask, the pixel-value $p$ indicates the spatial-temporal agreement degree over the 15 LULC products on the belonging of this pixel to the class represented by this mask.

---

**Algorithm 1** Description of the temporal combination process

1: **for** Each LULC class **do**
2:     Use the **AND** operator in the temporal combination
3:     Compute the number of obtained pixels $N$
4:     **if** $N \geq 1000$ **then**
5:         Keep using the **AND** operator in the temporal combination
6:     **else**
7:         Use the **MEAN** operator in the temporal combination
8:     **end if**
9: **end for**

---

**Algorithm 2** Description of the spatial combination process

1: **for** Each LULC class **do**
2:     Use **Rule 1** in the spatial combination
3:     Compute the number of obtained pixels $N$
4:     **if** $N \geq 1000$ **then**
5:         Keep using **Rule 1** in the spatial combination
6:     **else**
7:         Relax **Rule 1**
8:     **end if**
9: **end for**

---

**Table 4.** The rule set used to build the legend and define each LULC class in the TimeSpec4LULC dataset. P1 to P15: product 1 to 15. C1 to C29: class 1 to 29. The numbers from 0 to 220 correspond to class Ids in the original LULC products in Google Earth Engine. NU: Not Used, NA: Not Available, TC: Tree Cover, G: Gain, L: Loss, D: Datamask, TH: Tree Hight, TCC: Tree Canopy Cover, TCF: Tree-Cover Fraction, and SCF: Shrub-Cover Fraction)

| | P1 | P2 | P3 | P4 | P5 | P6 | P7 | P8 | P9 | P10 | P11 | P12 | P13 | P14 | P15 |
|---|---|---|---|---|---|---|---|---|---|---|---|---|---|---|---|
| C1 | 16 | 15 | NA | 7 | 11 | 60 | $TCC<10$ | 200 | 0 | 2 | $(TC<10)\cap(G=0)\cap(L=0)\cap(D\neq2)$ | $TH<1$ | $1\cup0$ | 0 | $Not(\geq1)$ |
| C2 | 16 | 15 | NA | 7 | 11 | 100 | $TCC<10$ | $200\cup150$ | 0 | 2 | $(TC<10)\cap(G=0)\cap(L=0)\cap(D\neq2)$ | $TH<1$ | $1\cup0$ | 0 | $Not(\geq1)$ |
| C3 | 10 | 10 | 1 | 6 | 6 | 30 | $TCC<10$ | 140 | NA | 2 | $(TC<10)\cap(G=0)\cap(L=0)\cap(D\neq2)$ | $TH<2$ | $1\cup0$ | 0 | $Not(\geq1)$ |
| C4 | 7 | 7 | 2 | NA | 5 | $20\cup(10<SCF<50)$ | $TCC<10$ | 150 | 0 | 2 | $(TC<10)\cap(G=0)\cap(L=0)\cap(D\neq2)$ | $TH<2$ | $1\cup0$ | 0 | $Not(\geq1)$ |
| C5 | 6 | 6 | 2 | NA | 5 | $20\cup(SCF>50)$ | $TCC<10$ | 130 | 0 | 2 | $(TC<10)\cap(G=0)\cap(L=0)\cap(D\neq2)$ | $TH<2$ | $1\cup0$ | 0 | $Not(\geq1)$ |
| C6 | NA | NA | NA | 4 | 4 | $4+(15<TCF<30)$ | $15<TCC<30$ | 60 | NA | 1 | $(15<TC<30)\cap(G=0)\cap(L=0)\cap(D\neq2)$ | $TH>2$ | $1\cup0$ | 0 | $Not(\geq1)$ |
| C7 | NA | NA | NA | 4 | 4 | $4+(40<TCF<60)$ | $40<TCC<60$ | 50 | NA | 1 | $(40<TC<60)\cap(G=0)\cap(L=0)\cap(D\neq2)$ | $TH>2$ | $1\cup0$ | 0 | $Not(\geq1)$ |
| C8 | 4 | 4 | 6 | 4 | 4 | $4+(TCF>60)$ | $TCC>60$ | 50 | NA | 1 | $(TC>60)\cap(G=0)\cap(L=0)\cap(D\neq2)$ | $TH>2$ | $1\cup0$ | 0 | $Not(\geq1)$ |
| C9 | NA | NA | NA | 3 | 3 | $3+(15<TCF<30)$ | $15<TCC<30$ | NA | NA | 1 | $(15<TC<30)\cap(G=0)\cap(L=0)\cap(D\neq2)$ | $TH>2$ | $1\cup0$ | 0 | $Not(\geq1)$ |
| C10 | NA | NA | NA | 3 | 3 | $3+(40<TCF<60)$ | $40<TCC<60$ | NA | NA | 1 | $(40<TC<60)\cap(G=0)\cap(L=0)\cap(D\neq2)$ | $TH>2$ | $1\cup0$ | 0 | $Not(\geq1)$ |
| C11 | 3 | 3 | 8 | 3 | 3 | $3+(TCF>60)$ | $TCC>60$ | NA | NA | 1 | $(TC>60)\cap(G=0)\cap(L=0)\cap(D\neq2)$ | $TH>2$ | $1\cup0$ | 0 | $Not(\geq1)$ |
| C12 | NA | NA | NA | 2 | 2 | $2+(15<TCF<30)$ | $15<TCC<30$ | 40 | NA | 1 | $(15<TC<30)\cap(G=0)\cap(L=0)\cap(D\neq2)$ | $TH>2$ | $1\cup0$ | 0 | $Not(\geq1)$ |
| C13 | NA | NA | NA | 2 | 2 | $2+(40<TCF<60)$ | $40<TCC<60$ | 40 | NA | 1 | $(40<TC<60)\cap(G=0)\cap(L=0)\cap(D\neq2)$ | $TH>2$ | $1\cup0$ | 0 | $Not(\geq1)$ |
| C14 | 2 | 2 | 5 | 2 | 2 | $2+(TCF>60)$ | $TCC>60$ | 40 | NA | 1 | $(TC>60)\cap(G=0)\cap(L=0)\cap(D\neq2)$ | $TH>2$ | $1\cup0$ | 0 | $Not(\geq1)$ |
| C15 | 9 | 9 | NA | 1 | 1 | $1+(15<TCF<30)$ | $15<TCC<30$ | 90 | NA | 1 | $(15<TC<30)\cap(G=0)\cap(L=0)\cap(D\neq2)$ | $TH>2$ | $1\cup0$ | 0 | $Not(\geq1)$ |
| C16 | 8 | 8 | 4 | 1 | 1 | $1+(40<TCF<60)$ | $40<TCC<60$ | 70 | NA | 1 | $(40<TC<60)\cap(G=0)\cap(L=0)\cap(D\neq2)$ | $TH>2$ | $1\cup0$ | 0 | $Not(\geq1)$ |
| C17 | 1 | 1 | 7 | 1 | 1 | $1+(TCF>60)$ | $TCC>60$ | 70 | NA | 1 | $(TC>60)\cap(G=0)\cap(L=0)\cap(D\neq2)$ | $TH>2$ | $1\cup0$ | 0 | $Not(\geq1)$ |
| C18 | 11 | 11 | NA | NA | NA | 90 | $TCC>10$ | 170 | NA | NA | $(TC>10)\cap(G=0)\cap(L=0)\cup(D=2)$ | $TH>2$ | $2\cup3$ | 1 | $Not(\geq1)$ |
| C19 | 11 | 11 | NA | NA | NA | 90 | $TCC>10$ | $a.160\cup180$ $b.Not(170)$ | NA | NA | $(TC>10)\cap(G=0)\cap(L=0)\cup(D=2)$ | $TH>2$ | $2\cup3$ | 1 | $Not(\geq1)$ |
| C20 | 11 | 11 | NA | NA | NA | 90 | $TCC<10$ | $160\cup170$ $\cup180$ | NA | NA | $(TC<10)\cap(G=0)\cap(L=0)\cup(D=2)$ | $TH<2$ | $2\cup3$ | 1 | $Not(\geq1)$ |
| C21 | 17 | 0 | 0 | 0 | 0 | 200 | NA | 210 | NA | 3 | NA | NA | 3 | 1 | $Not(\geq1)$ |
| C22 | 17 | 0 | 0 | 0 | 0 | 80 | NA | 210 | NA | 3 | NA | NA | 3 | 1 | $Not(\geq1)$ |
| C23 | 15 | NA | NA | NA | 10 | 70 | NA | 220 | NA | NA | NA | NA | $1\cup0$ | 0 | $Not(\geq1)$ |
| C24 | 12 | 12 | $3\cup1$ | $5\cup6$ | $7\cup8$ | 40 | NA | $11\cup14$ | $1\cup2\cup3$ $\cup4\cup5$ | NA | NA | NA | $2\cup3$ | $0\cup4\cup5$ $\cup8\cup10$ | $Not(\geq1)$ |
| C25 | 12 | 12 | 1 | 6 | 7 | 40 | NA | 11 | $1\cup2$ | NA | NA | NA | $1\cup0$ | 0 | $Not(\geq1)$ |
| C26 | 12 | 12 | 1 | 6 | 7 | 40 | NA | 14 | $3\cup4\cup5$ | NA | NA | NA | $1\cup0$ | 0 | $Not(\geq1)$ |
| C27 | 12 | 12 | 3 | 5 | 8 | 40 | NA | 11 | $1\cup2$ | NA | NA | NA | $1\cup0$ | 0 | $Not(\geq1)$ |
| C28 | 12 | 12 | 3 | 5 | 8 | 40 | NA | 14 | $3\cup4\cup5$ | NA | NA | NA | $1\cup0$ | 0 | $Not(\geq1)$ |
| C29 | 13 | 13 | 10 | 8 | 9 | 50 | NA | 190 | NA | NA | NA | NA | $1\cup0$ | 0 | NU |

### 2.1.4 Re-sampling and selection of agreement threshold

The final mask of each LULC class maintained the spatial resolution of the last aggregated LULC product P15 at 30 m resolution. The 30-meter resolution LULC consensus was re-sampled with MODIS resolution (approximately equal to 500 m) using the spatial MEAN reducer. This 500 m average consensus was used to explore different agreement thresholds $\theta$ for each LULC class. We used $\theta = 1$, when the number of retrieved 500-meter pixels is greater than 1000, which means that the 15 LULC products totally agree on the class type of these pixels. Otherwise, we decreased the threshold $\theta$ by 0.05 until we reached at least 1000 pixels (Algorithm 3). Table 7 provides the number of pixels obtained with each agreement threshold. In any case, our dataset provides as metadata, the agreement percentage at pixel level, so that the user can control the desired agreement threshold and subsequent sample size. To ensure collecting at least 1000 pixels from each class, the lowest pixel-agreement threshold used is $\theta = 0.80$ (Table 7).

After performing, for each LULC class, the spatial-temporal agreement, the re-projection, and the selection of the agreement threshold, we combined the final class masks of all the 29 LULC classes to generate one global LULC map describing their distribution (Fig. 2). This figure shows in which place of the world the 29 LULC classes are more stable in time, and the 15 LULC products are more compliant, since the number of the collected pixels in each class is affected by the temporal consistency of the 29 LULC classes and the spatial consistency over the 15 LULC products. To illustrate all the steps of the spatial-temporal agreement process across the 15 global LULC products, we provide an example explaining the generation of the final mask for the class Dense Evergreen Broadleaf Forests (Fig. 3).

**Table 5.** Description of the Temporal-Spatial Combination of the 15 global LULC products (P1:P15) masks to build a consensus image for each LULC class. ([1]: Inter-annual combination used in all products except in P13, where we first calculated the inter-annual Mean and then transformed it into a water no-water binary mask)

| Class Id(s) | LULC class(s) | Spatial Combination | Temporal Combination [1] |
|---|---|---|---|
| C1 | Barren Lands | Rule 1: Mean(P1 : P12) * P13 * P14 * P15 | Operator 1: AND |
| C2 | Moss and Lichen Lands | Rule 2: Mean(P1 : P5, P7 : P12) * P6 * P13 * P14 * P15 | Operator 2: MEAN |
| C3 | Grasslands | Rule 1: Mean(P1 : P12) * P13 * P14 * P15 | Operator 1: AND |
| C4 | Open Shrublands | Rule 1: Mean(P1 : P12) * P13 * P14 * P15 | Operator 1: AND |
| C5 | Closed Shrublands | Rule 1: Mean(P1 : P12) * P13 * P14 * P15 | Operator 2: MEAN |
| C6:C17 | Forests | Rule 1: Mean(P1 : P12) * P13 * P14 * P15 | Operator 2: MEAN |
| C18 | Mangrove Wetlands | Rule 3: Mean(P1 : P7, P9 : P14) * P8 * P15 | Operator 2: MEAN |
| C19 | Swamp Wetlands | Rule 4: Mean(P1 : P8.a, P9 : P12) * P8.b * P15 | Operator 2: MEAN |
| C20 | Marshland Wetlands | Rule 5: Mean(P1 : P6, P8 : P10, P13 : P14) * P7 * P11 * P12 * P15 | Operator 2: MEAN |
| C21:C22 | Water Bodies | Rule 1: Mean(P1 : P12) * P13 * P14 * P15 | Operator 1: AND |
| C23 | Permanent Snow | Rule 1: Mean(P1 : P12) * P13 * P14 * P15 | Operator 1: AND |
| C24 | Croplands Flooded with seasonal water | Rule 6: Mean(P1 : P12) * (P13 OR P14) * P15 | Operator 1: AND |
| C25:C26 | Cereal Croplands | Rule 1: Mean(P1 : P12) * P13 * P14 * P15 | Operator 1: AND |
| C27:C28 | Broadleaf Croplands | Rule 1: Mean(P1 : P12) * P13 * P14 * P15 | Operator 1: AND |
| C29 | Urban and Built-Up Areas | Rule 1: Mean(P1 : P12) * P13 * P14 * P15 | Operator 1: AND |

---

**Algorithm 3** Selection of agreement threshold for each LULC class

---

1: **for** Each LULC class **do**

2:     Set the agreement threshold of the LULC class mask to $\theta = 1$

3:     Compute the number of obtained pixels $N$

4:     **while** $N < 1000$ **do**

5:         Reduce the agreement threshold ($\theta \leftarrow \theta - 0.05$)

6:         Compute the number of obtained pixels $N$

7:     **end while**

8:     Generate the final class mask for the LULC class using the agreement threshold $\theta$

9: **end for**

---

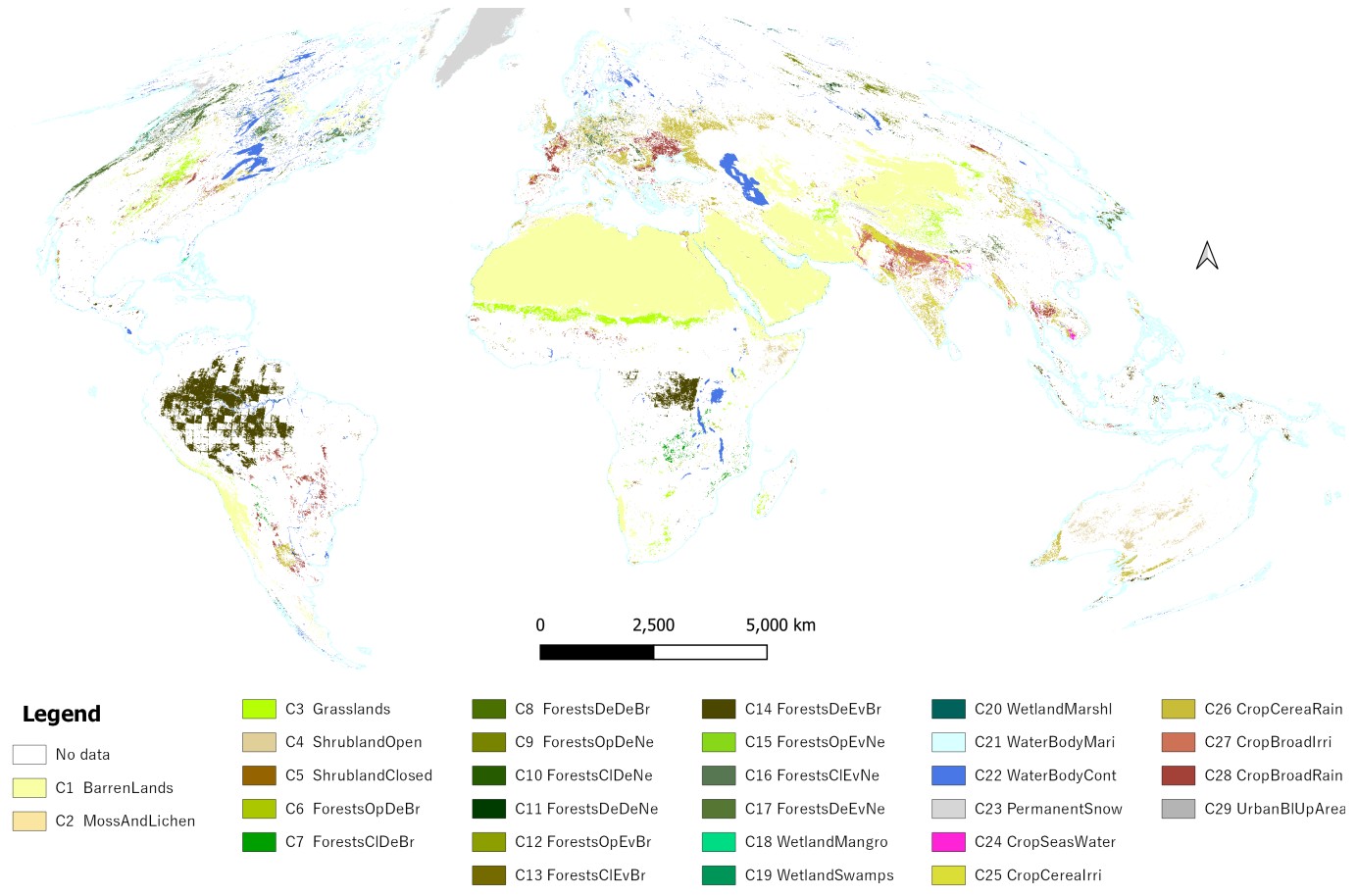

**Figure 2.** Distribution of the number of covered countries (Food and Agricultural Organization's Global Administrative Unit Layers 2015 GAUL-ADM0) over the 29 LULC classes. This map combines all the final LULC class masks that were generated from the process of spatial-temporal agreement across the 15 global LULC products available in GEE. In the Map's legend we are presenting the short names of the LULC classes (Their corresponding full names are presented in Table 3).

## 2.2 Extracting times series of spectral data for 29 LULC classes globally

To extract the 22-year monthly time series of the seven 500-meter MODIS spectral bands for each of the 29 LULC classes throughout the entire world, we followed four steps (Fig. 4): 1) spatial-temporal filtering of Terra and Aqua data based on Quality Assessment flags, 2) aggregating the original 8-day Terra and Aqua data into monthly composites, 3) merging the two monthly time series into a Terra+Aqua Combined time series, and 4) data extraction and archiving (Fig. 6).

### 2.2.1 Spatial-temporal filtering of Terra and Aqua data based on quality assessment flags

MODIS sensor has high temporal coverage, ensured by Terra and Aqua satellites revisit frequencies, and also spectral and spatial features that are highly suitable for LULC mapping and change detection (García-Mora et al., 2012; Xiong et al.,

2017). Thus, we used two MODIS products MOD09A1 (Ter) and MYD09A1 (Aqu) that estimate the 8-day surface spectral
reflectance for the seven 500-meter bands from Terra and Aqua, respectively.

The quality of any time series of satellite imagery is affected by the internal malfunction of satellite sensors, atmospheric
(e.g., clouds, shadows, cirrus, etc.) or land (e.g., floods, snow, fires, etc.) conditions. In addition to the spectral bands, MODIS
products provide 'Quality Assessment' (QA) flags as metadata bands to allow the user to filter out spectral values affected by
disruptive conditions. Therefore, all QA flags were used to remove noise, spurious values, and outliers in the image collection.
"MODLAND QA" flags (bits 0-1) were used to only select pixel values produced at 'ideal quality'.

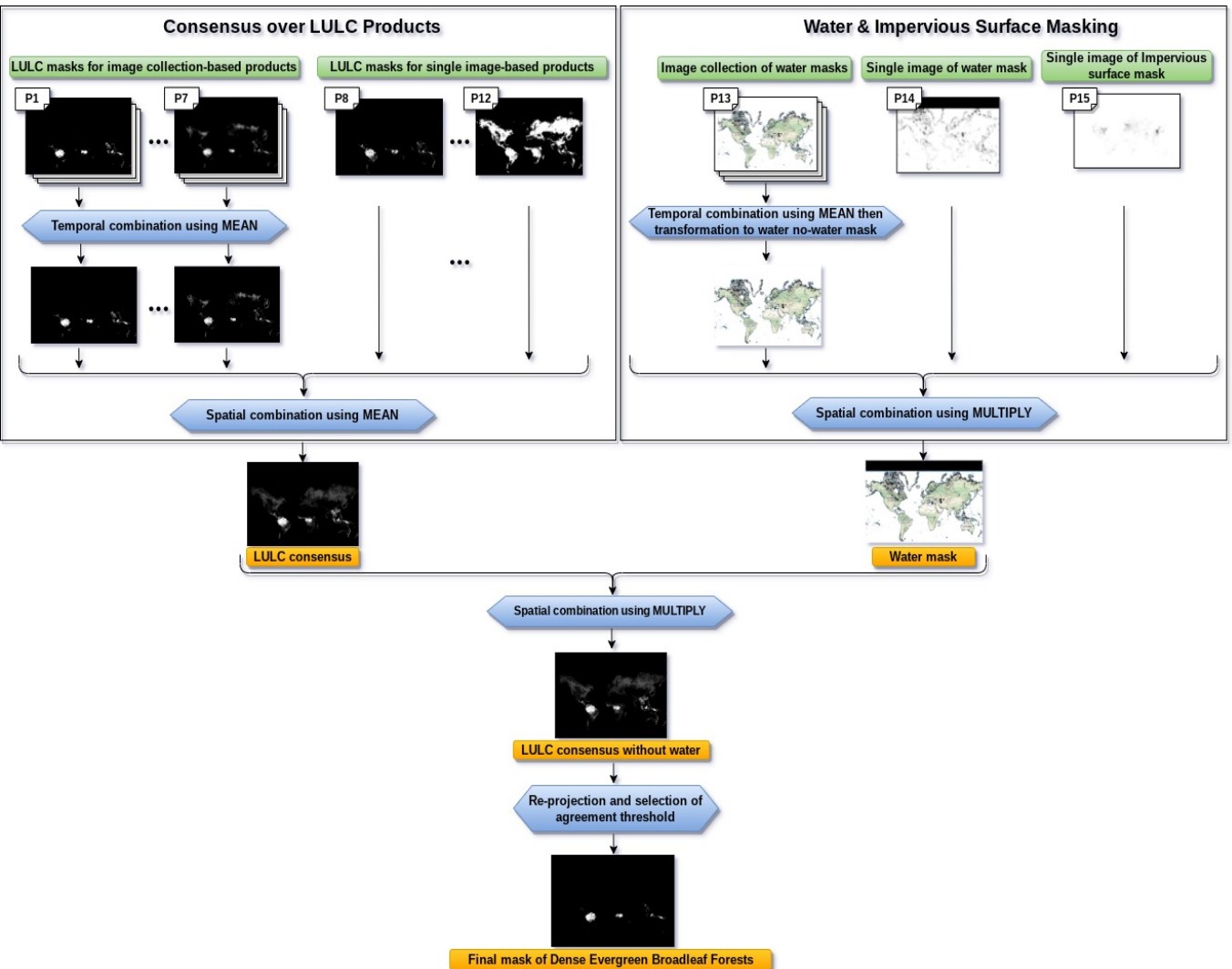

**Figure 3.** Example of the final mask creation process for the Dense Evergreen Broadleaf Forests LULC class produced through the spatial-temporal agreement over the 15 global LULC products available in GEE.

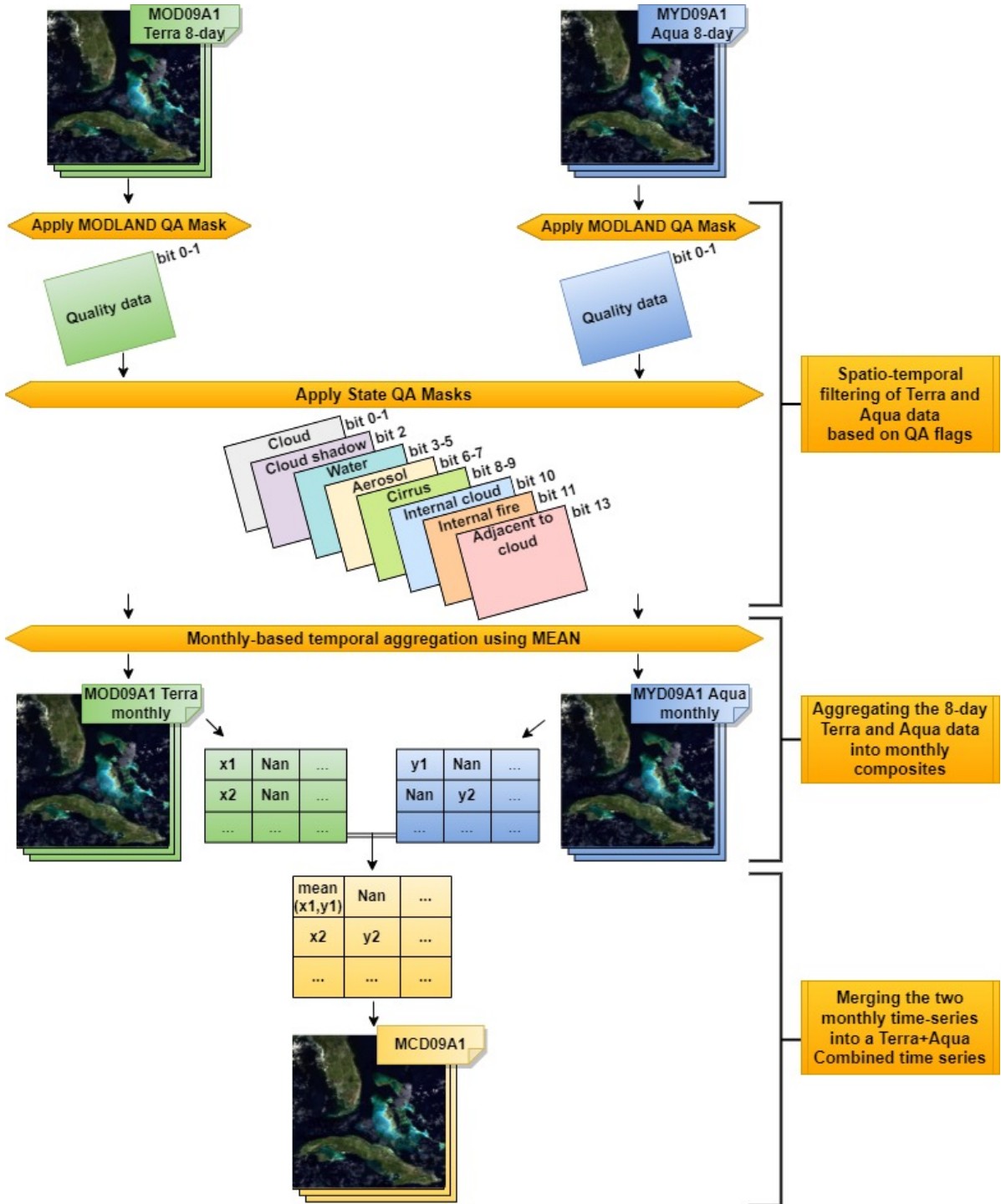

**Figure 4.** Description of the spatial-temporal filtering of Terra and Aqua, their aggregation into monthly composites, and their merging into Terra+Aqua Combined time series. This process aims to filter out spectral values affected by disruptive conditions, and to reduce the number of gaps in the multi-spectral time series for the 29 LULC classes.

Then, "State QA" flags were used to mask out clouds (bits 0-1), internal clouds (bit 10), pixels adjacent to clouds (bit 13), cirrus (bits 8-9), cloud shadows (bit 2), high aerosol quantities (bits 6-7), and internal fires (bit 11). The water flag (bits 3-5) was used to mask out water pixels in all terrestrial systems, but not in the terrestrial systems of Permanent Snow, and in Croplands Flooded with Seasonal Water to avoid unrealistic data loss.

### 2.2.2 Aggregating the original 8-day Terra and Aqua data into monthly composites

Filtering the MODIS Terra and Aqua data records produced many missing-values in their 8-day time series. To overcome this issue and further reduce the presence of noise in our dataset, the original 8-day time series were aggregated into monthly composites by computing the mean over the observations of each month. Indeed, despite reducing the temporal resolution from 8-day to monthly composites shortened the time series size, it generated two datasets with less missing values and clear monthly patterns, which are more intuitive to track LULC dynamics than the 8-day patterns.

### 2.2.3 Merging the two monthly time series into a Terra+Aqua combined time series

Terra satellite daily orbits above the Earth's surface from north to south in the morning at around 10:30 local time, while Aqua orbits in the opposite direction in the afternoon at around 13:30. Having two opportunities per day at each location increases the chances of capturing an image under good atmospheric conditions. To further reduce the number of missing values in our dataset, we merged the monthly time series provided by these two satellites into a Terra+Aqua Combined time series. That is, for each pixel, band, and month, when both Terra and Aqua had values, we used the mean between them; when one satellite had a missing value, we used the available one; and when both of them had missing values, the combined value remains missing. Since Aqua was launched three years later (in 2002) after Terra had been launched, the acquisition time of our dataset is: (1) from 05-03-2000 to 04-07-2002 using Terra time series, and (2) from 04-07-2002 to 19-12-2021 using Terra+Aqua time series.

### 2.2.4 Extracting and archiving the dataset

One of the main advantages of our dataset is its global scale characteristic since all the LULC data were extracted globally from all the regions over the world. The data exportation process was performed in two steps (Fig. 6): (1) We exported the metadata of all the pixels generated by the consensus. Then, (2) we exported their corresponding time series data. Detailed descriptions and discussions about each step are provided as follows:

1. From each LULC class mask, we first exported the metadata of all the available pixels in one file. However, for the class masks having more than 1 million pixels (Barren Lands, Water Bodies, Permanent Snow, Grasslands, Open Shrublands, and Dense Evergreen Broadleaf Forests) we only exported the metadata of 500,000 pixels randomly selected over the globe because of the memory limitations in GEE (Table 7). The exported metadata includes the coordinates of the pixel center, and the percentage of agreement over the 15 LULC products. To take into account all the differences across the globe and thinking of regional interests that some users may have, we used the Food and Agricultural Organization's Global Administrative Unit Layers 2015 (FAO GAUL) product, available in GEE, to provide for each pixel the ADM0-

CODE obtained from the Country Boundaries (GAU, a) of FAO GAUL (i.e., countries), and the ADM1-CODE obtained from the First-Level Administrative Units (GAU, b) of FAO GAUL (e.g., departments, states, provinces). Further, to provide the user with extra metadata that could be used to filter time series according to different levels of human intervention on each pixel, the average GHM index was included. The GHM index was derived from the Global Human Modification dataset (CSP gHM) (Kennedy et al., 2019) available in GEE, which provides a cumulative measure of human modification of terrestrial lands. Then, it was projected to MODIS resolution using the spatial mean reducer to generate the average GHM index.

2. After exporting the metadata, we accessed the coordinates of each LULC class to download their time series data for the 7 spectral bands (Table 6). Each time series data contain 262 observations covering almost 22 years (i.e., from 2000 to the end of 2021). In order to optimize the exportation process, for each LULC class, the 262 observations corresponding to the 262 months were exported separately in 262 parallel requests. In each request, we exported 7 values corresponding to the 7 spectral bands for all the LULC class related pixels.

**Table 6.** Description of the seven spectral bands of Modis sensor.

| Band Id | Band Name | Wavelength | Description |
|---------|-----------|------------|-------------|
| B1 | MCD09A1_B1 | 620-670 nm | Surface reflectance for band 1 |
| B2 | MCD09A1_B2 | 841-876 nm | Surface reflectance for band 2 |
| B3 | MCD09A1_B3 | 459-479 nm | Surface reflectance for band 3 |
| B4 | MCD09A1_B4 | 545-565 nm | Surface reflectance for band 4 |
| B5 | MCD09A1_B5 | 1230-1250 nm | Surface reflectance for band 5 |
| B6 | MCD09A1_B6 | 1628-1652 nm | Surface reflectance for band 6 |
| B7 | MCD09A1_B7 | 2105-2155 nm | Surface reflectance for band 7 |

The exported data generated highly imbalanced LULC classes obviously due to the differences in their spatial distributions. Thus, to facilitate the exploration of the dataset, we also provided a balanced dataset ready to train machine learning models. The balanced subset of TimeSpec4LULC provides the time series data for 1000 pixels from each class since the smallest LULC class contains 1194 pixels (Table 7). The selection of these 1000 samples from each class was performed using Algorithm 4 such that they are evenly distributed in the globe and representative for the world. In Figure 5, we provide the distribution of the 1000 pixels selected from the class Marine Water Bodies.

The provided metadata can also be used in case the user wants to export future time series observations for the coming months. In this context, the user needs to make use of the ADM0-CODE and the ADM1-CODE to access the coordinates of any region in the world included in the consensus. Then, upload these coordinates to GEE to export the time series data of the desired range of time.

**Algorithm 4** Selection of evenly distributed pixels for one LULC class

1: Load the metadata of one LULC class.

2: Remove the coordinates of the empty pixels based on the original data version.

3: Put the cleaned coordinates of the LULC class in $Data\_Points$.

4: Choose a random point from $Data\_Points$ and add it to $Selected\_Points$.

5: **while** $|Selected\_Points| < 1000$ **do**

6:    **for** $Point$ **in** $Data\_Points$ **do**

7:       Compute the distance between $Point$ and $Selected\_Points$.

8:       Get the minimum distance and add it to $Distance\_To\_Closest\_Points$.

9:    **end for**

10:    Get the index of the maximum distance:

$$\arg\max(Distance\_To\_Closest\_Points)$$

11:    Add the corresponding $Point$ to $Selected\_Points$.

12: **end while**

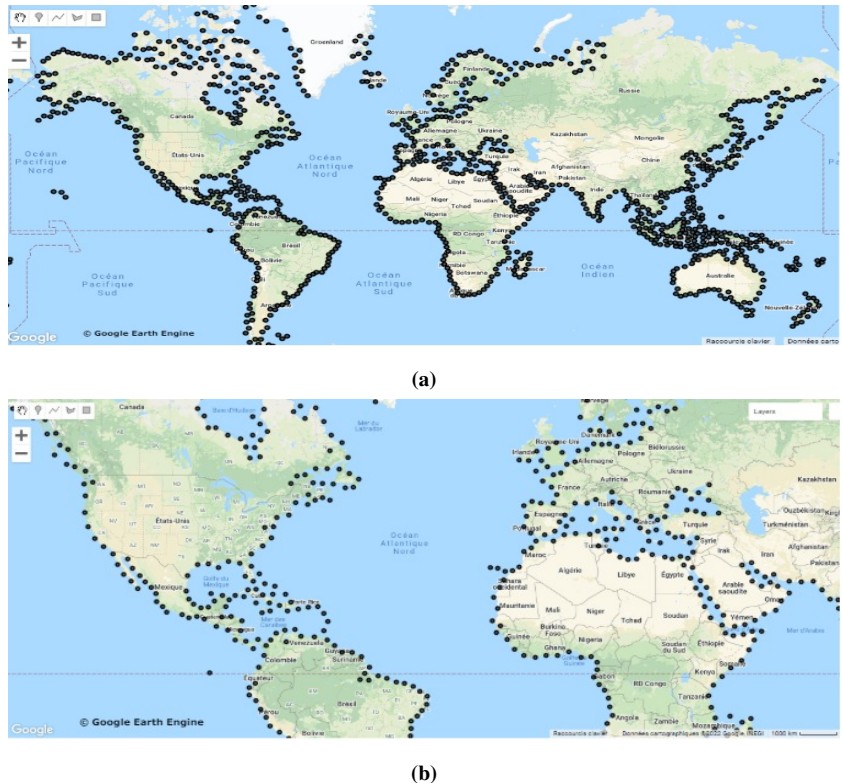

(a)

(b)

**Figure 5.** Distribution of the 1000 points selected by Algorithm 4 for the class Marine Water Bodies: (a) global view, (b) zoom-in view.

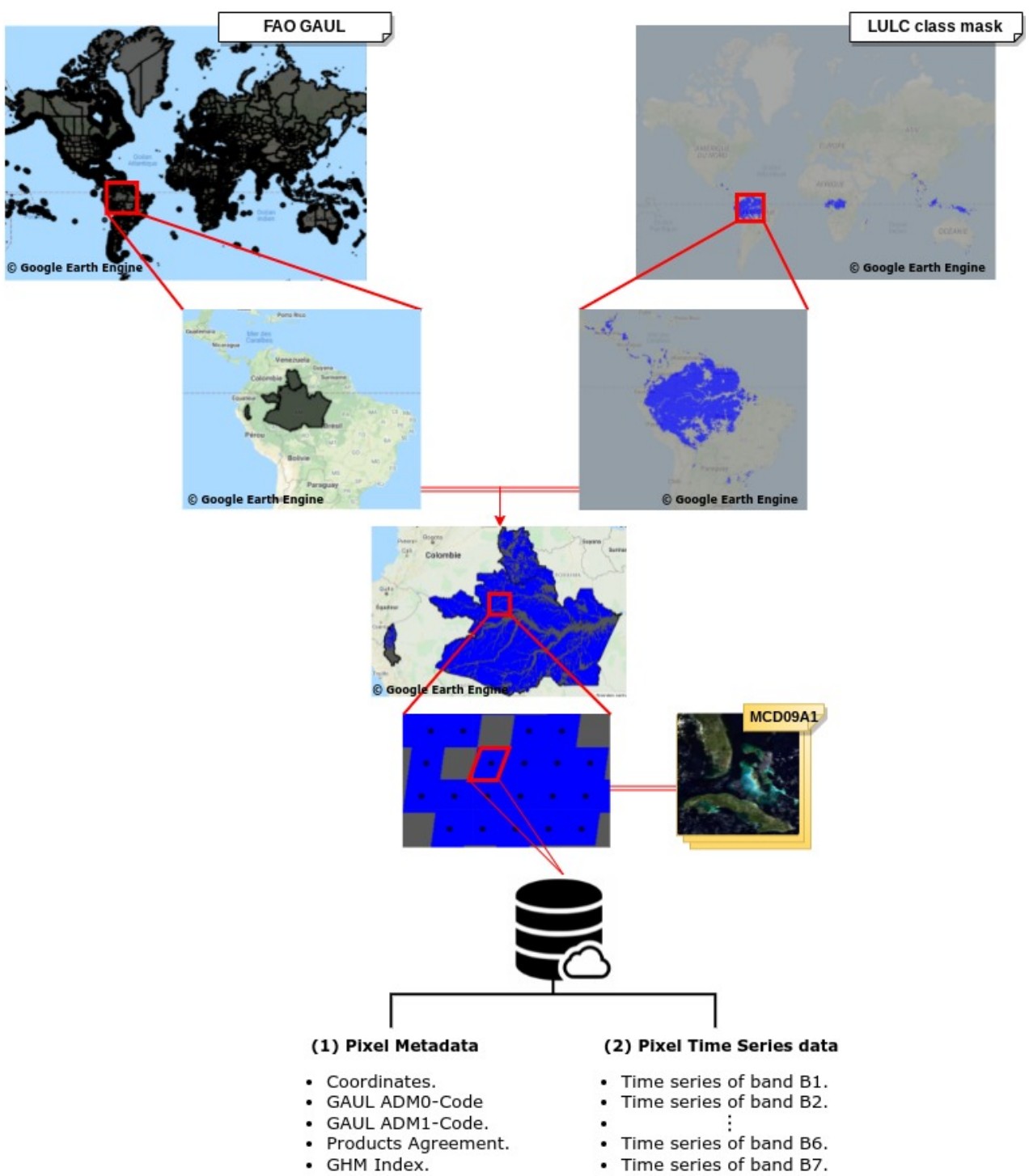

**Figure 6.** Description of the data extraction process for the LULC class Dense Evergreen Broadleaf Forests from all the world's partitions (GAUL-ADM0) and (GAUL-ADM1) where this class is available.

**Table 7.** Sensitivity analysis of the number of pixels with respect to different values of agreement thresholds along with the final number of collected pixels at the selected threshold. When the number of pixels at threshold 1 exceeds 1 million, we collect 500,000 random pixels. Otherwise, we decrease the threshold by 0.05 until we obtain at least 1000 pixels (Algorithm 3).

| Class Id | Class short name | Agreement thresholds | | | | | Collected pixels | Selected threshold |
|---|---|---|---|---|---|---|---|---|
| | | 0.80 | 0.85 | 0.90 | 0.95 | 1 | | |
| C1 | BarrenLands | 85,293,945 | 83,484,114 | 81,157,460 | 73,495,569 | 65,332,858 | 500,000 | 1 |
| C2 | MossAndLichen | 646,305 | 482,619 | 287,757 | 134,549 | 2807 | 2807 | 1 |
| C3 | Grasslands | 55,588,334 | 34,749,935 | 21,729,176 | 4,082,093 | 1,032,092 | 500,000 | 1 |
| C4 | ShrublandOpen | 32,024,725 | 21,664,056 | 14,594,193 | 2,117,778 | 223,062 | 223,062 | 1 |
| C5 | ShrublandClosed | 549,792 | 128,113 | 38,656 | 2985 | 9 | 2985 | 0.95 |
| C6 | ForestsOpDeBr | 130,123 | 7034 | 4 | 0 | 0 | 7034 | 0.85 |
| C7 | ForestsClDeBr | 486,196 | 41,869 | 494 | 0 | 0 | 41,869 | 0.85 |
| C8 | ForestsDeDeBr | 6,646,105 | 4,765,433 | 2,993,393 | 387,276 | 2240 | 2240 | 1 |
| C9 | ForestsOpDeNe | 1402 | 28 | 0 | 0 | 0 | 1402 | 0.80 |
| C10 | ForestsClDeNe | 71,446 | 1348 | 0 | 0 | 0 | 1348 | 0.85 |
| C11 | ForestsDeDeNe | 1,109,793 | 703,062 | 242,614 | 10,979 | 0 | 10,979 | 0.95 |
| C12 | ForestsOpEvBr | 2719 | 86 | 0 | 0 | 0 | 2719 | 0.80 |
| C13 | ForestsClEvBr | 58,552 | 3322 | 149 | 1 | 0 | 3322 | 0.85 |
| C14 | ForestsDeEvBr | 49,150,065 | 45,678,189 | 40,445,318 | 32,048,990 | 3,000,060 | 500,000 | 1 |
| C15 | ForestsOpEvNe | 2735 | 10 | 0 | 0 | 0 | 2735 | 0.80 |
| C16 | ForestsClEvNe | 154,341 | 4332 | 26 | 0 | 0 | 4332 | 0.85 |
| C17 | ForestsDeEvNe | 6,987,918 | 4,562,614 | 1,966,655 | 558,406 | 362 | 558,406 | 0.95 |
| C18 | WetlandMangro | 14,095 | 4750 | 716 | 78 | 0 | 4750 | 0.85 |
| C19 | WetlandSwamps | 8453 | 1194 | 100 | 7 | 0 | 1194 | 0.85 |
| C20 | WetlandMarshl | 18,748 | 9491 | 4500 | 1405 | 80 | 1405 | 0.95 |
| C21 | WaterBodyMari | 47,953,196 | 46,869,483 | 40,323,857 | 39,200,046 | 35,848,199 | 500,000 | 1 |
| C22 | WaterBodyCont | 47,541,101 | 45,792,728 | 6,016,114 | 5,630,082 | 4,789,580 | 500,000 | 1 |
| C23 | PermanentSnow | 7,593,382 | 7,540,486 | 7,469,482 | 7,354,210 | 6,827,318 | 500,000 | 1 |
| C24 | CropSeasWater | 233,404 | 190,947 | 134,486 | 97,732 | 38,642 | 38,642 | 1 |
| C25 | CropCerealrri | 6,559,822 | 4,949,682 | 1,392,245 | 1,005,469 | 405,340 | 405,340 | 1 |
| C26 | CropCereaRain | 17,025,686 | 13,632,125 | 6,334,106 | 3,693,354 | 848,583 | 848,583 | 1 |
| C27 | CropBroadIrri | 2,977,417 | 2,349,114 | 1,099,282 | 896,775 | 392,630 | 392,630 | 1 |
| C28 | CropBroadRain | 6,965,150 | 5,686,144 | 2,596,559 | 1,561,992 | 359,674 | 359,674 | 1 |
| C29 | UrbanBlUpArea | 1,832,276 | 1,178,905 | 704,481 | 501,219 | 159,073 | 159,073 | 1 |
| | Total number of collected pixels | | | | | | 6,076,531 | |

 **3   Data**

To organize and assess the quality of the extracted global data for all the 29 LULC classes, we first present the description of the dataset structure, then we evaluate the quality of its annotation process.

**3.1   Description of the data structure**

TimeSpec4LULC dataset is hosted by Zenodo repository (Khaldi et al., 2022). It contains two datasets: the original dataset
"TimeSpec4LULC_Original_data.zip", and the balanced subset of the original dataset "TimeSpec4LULC_Balanced_data.zip". The structure of TimeSpec4LULC is organized as follows (Figure 7):

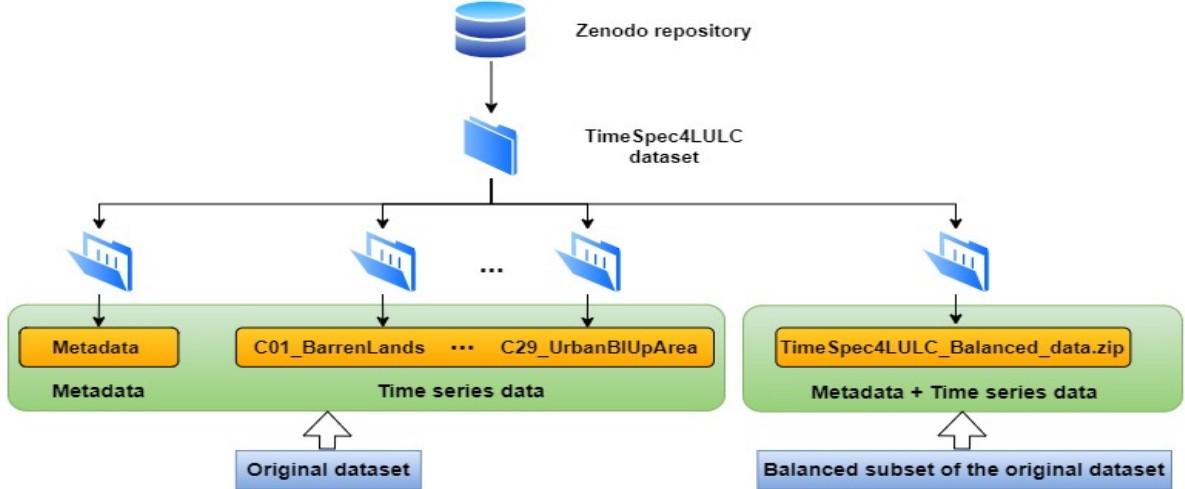

**Figure 7.** Dataset structure.

– The original dataset contains 30 folders, namely "Metadata", and 29 folder corresponding to the 29 LULC classes. The folder "Metadata" holds 29 different CSV files named on behalf of the 29 LULC classes. The naming of each file follows the structure "ClassId_metadata.csv". For instance, the metadata CSV file for the Barren Lands class is named
"C01_metadata.csv". Each CSV file holds the metadata of all the pixels generated by the consensus limited to 500,000 for classes that exceed 1 million at agreement threshold 1.

The remaining 29 folders contain the time series data for the 29 LULC classes. Each folder has the form "ClassId_ClassShortName", and holds 262 CSV files corresponding to the 262 months. For example, the CSV file for the Barren Lands class for the last month is named "C01_261.csv". Inside each CSV file, we provide the seven values of
the spectral bands as well as the coordinates for all the LULC class related pixels.

A clear description of the metedata folder along with an example of the time series data for Barren Lands is presented in Fig. 8.

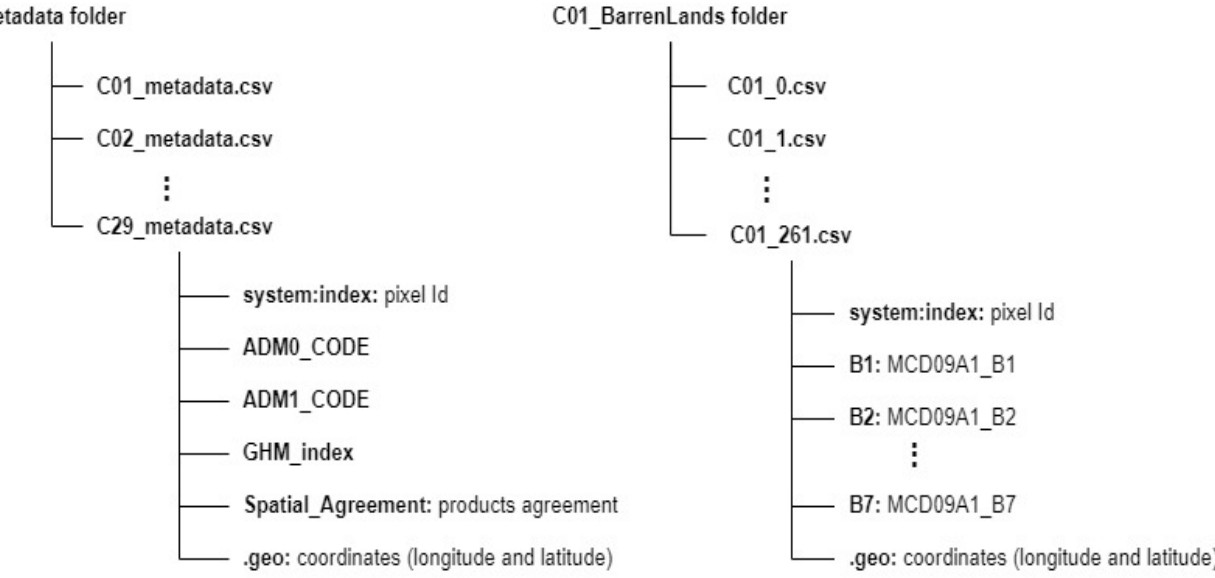

**Figure 8.** Data structure of the metadata folder (left), and the time series data folder (right) for the class Barren Lands in the original dataset.

– The balanced subset of the original dataset holds the metadata and the time series data for 1000 pixels per class representative of the globe selected by Algorithm 4. It contains 29 different JSON files following the names of the 29 LULC classes. The naming of each file follows the structure "ClassId_ClassShortName.json". For instance, the JSON file for the Barren Lands class is named "C01_BarrenLands.json".

Each JSON file "Class_File" is a dictionary containing the short name of the LULC class "Class_Name", the Id of the class "Class_Id", and a list of all the relative pixels "Pixels" (for more information about the LULC classes short names, see Table 3). Each element of the list "Pixels" is a dictionary holding the Id of the pixel "Pixel_Id", the class of the pixel "Pixel_Label", the metadata of the pixel "Pixel_Metadata", and the 7 time series of the pixel "Pixel_TS".

The variable "Pixel_Metadata" contains the geometry and coordinates (longitude and latitude) of the pixel center following the GEE format ".geo", the GAUL country code "ADM0_Code", the GAUL first level administrative unit code "ADM1_Code", the average of the global human modification index "GHM_Index", the agreement percentage over the 15 LULC products "Products_Agreement_Percentage", and a dictionary carrying the temporal availability percentage for each band "Temporal_Availability_Percentage" (i.e., percentage of non-missing data per band from B1 to B7).

The variable "Pixel_TS" is a dictionary that holds the names and the time series values of the seven spectral bands (from "MCD09A1_B1" to "MCD09A1_B7") of size 262. A clear description of the JSON class file is presented in Fig. 9.

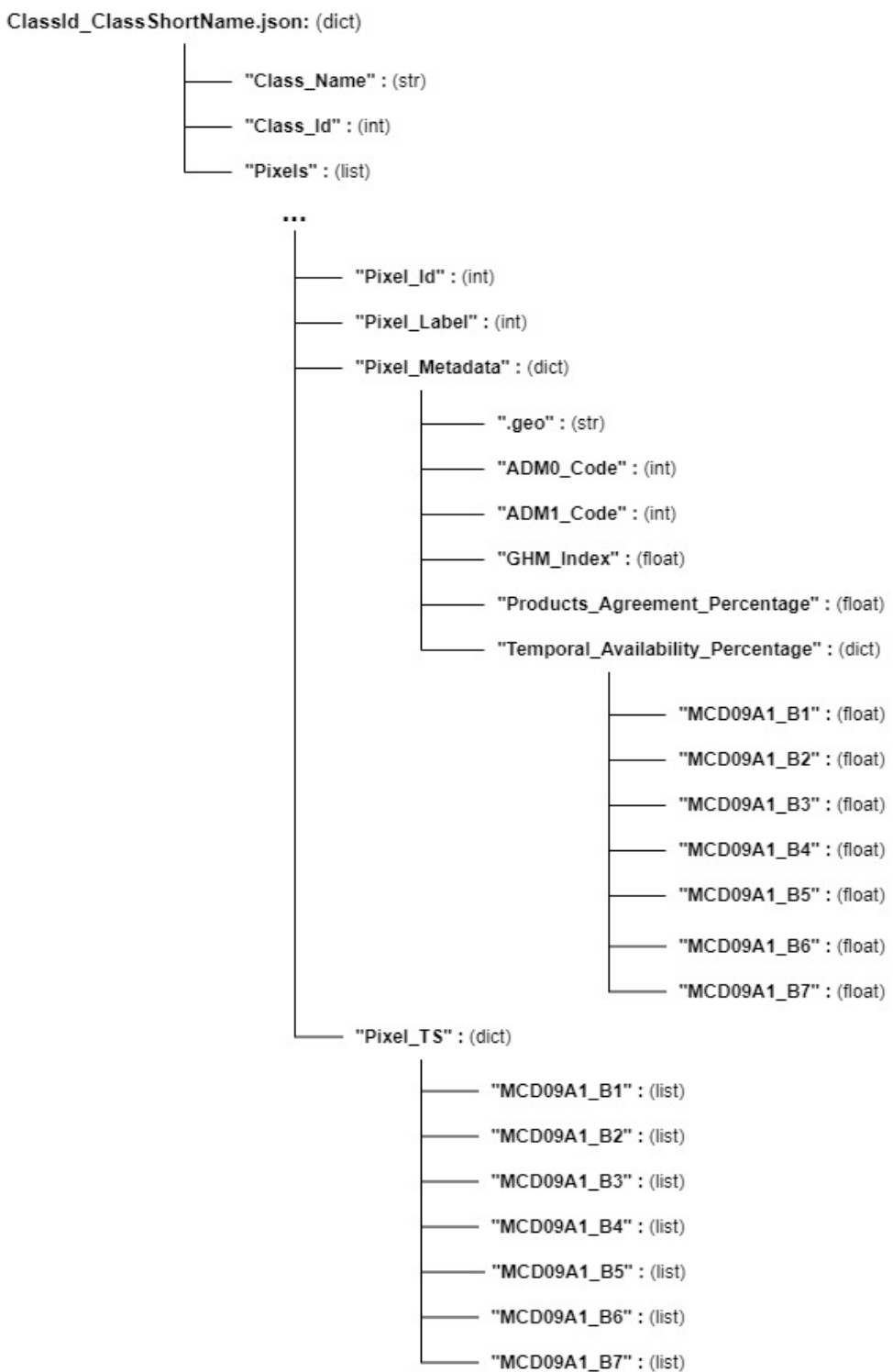

**Figure 9.** Data structure of an LULC class JSON file for the balanced subset of the original dataset.

## 3.2 Data quality control

The quality of the dataset annotation was assessed and validated visually by two co-author experts using two very high res-
olution imagery (< 1m/pixel) sources, namely Google Earth and Bing Maps imagery. The assessment process includes three
stages:

- First, a set of 100 samples is carefully selected from each class following the maximum distance criteria described in
  Algorithm 4. That is, depending on the overall size of each LULC class, 100 evenly distributed pixels over the globe
  were selected. Fig. 10 shows the distribution of the 2,900 selected pixels.

- Second, the class of each pixel of the $29\times100$ samples is identified visually by the expert eye following the next rule. We
  consider as ground truth the dominant LULC class, such LULC class occupies at least 70% of the pixel. The presence of
  up to 30% of features of other different LULC classes within the dominate class are ignored.

- Once the validated LULC classification matrix was obtained (Table 8), the F1 score was calculated for all the LULC
  levels (from L0 to L5). We used F1 score because it evaluates the balance between *precision* and *recall*. Where, precision
  indicates how accurate the annotation process is in predicting true positives, and (2) the Recall, also called sensitivity,
  indicates how many actual positives were predicted as true positives (Eq.3).

$$F1-score = 2 \times \frac{Precision \times Recall}{Precision + Recall} \tag{1}$$

$$Precision = \frac{TruePositive}{TruePositive + FalsePositive} \tag{2}$$

$$Recall = \frac{TruePositive}{TruePositive + FalseNegative} \tag{3}$$

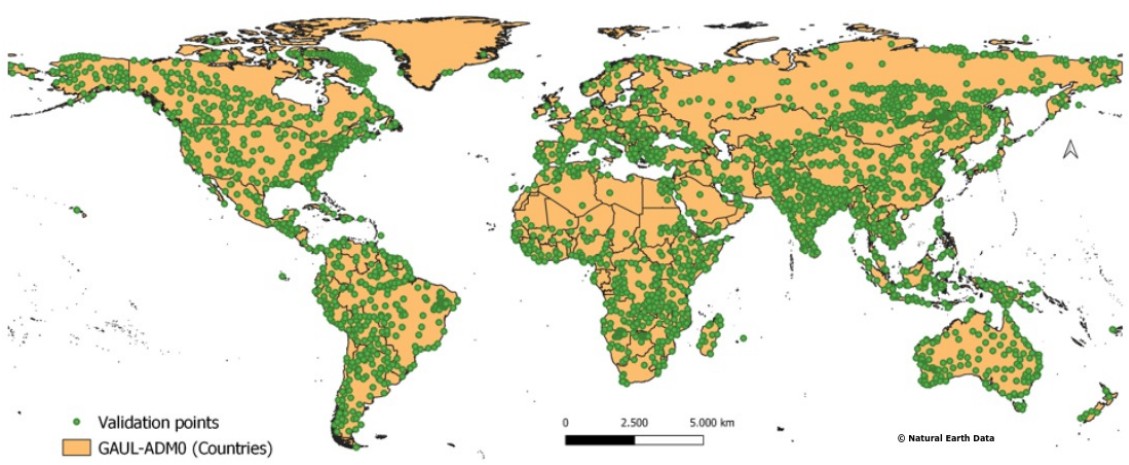

**Figure 10.** Global distribution of the selected 2,900 pixels to perform the quality control over all the 29 LULC classes.

As it can be observed from Table 8, as we go up from level L0 to level L5 the obtained *F1 score* decreases from 96% to 87% mainly due to the classification of Forests, Grasslands, Open Shrublands, Water Bodies, and Croplands flooded with seasonal water. Typically, the obtained *F1 score* of each class is independent of the selected agreement threshold. In some classes, even if the agreement threshold is equal to 1, the *F1 score* is low compared to other classes with small agreement threshold. For instance, the agreement threshold of Grasslands and Open Deciduous Needleleaf Forests is equal to 1 and 0,80, respectively. However, the *F1 score* of Grasslands is lower (0,68) than the *F1 score* of Open Deciduous Needleleaf Forests (0,90).

## 4 Results and discussions

The total number of collected time series (pixels) in all the 29 LULC classes is 6,076,531 which is large enough to build high quality DL models (Table 7). This number covers the 29 LULC classes in unbalanced way due to two reasons: (1) the global abundance of each class, and (2) the choice of the agreement threshold. We provide, in Table 7, the variation of the number of pixels with respect to different values of agreement thresholds. It can be noticed that as we decrease the agreement threshold, the number of pixels increases. Table 7 highlights also the classes that reduced the consensus F1-score (with selected threshold less than 1) which are: the three Wetlands classes, Closed Shrublands, and all Forests classes except Dense Deciduous Broadleaf and Dense Evergreen Broadleaf.

In 15 LULC classes, the number of collected time series, at agreement threshold 1, is at least 2,240 per class. This means that the 15 LULC products are 100% compliant with regard to the nature of these classes. Thus, these classes have enough pure spectral information, describing their behavior over time, to train DL models with very high accuracy. However, in the remaining 14 LULC classes, the number of time series, collected at agreement threshold 100%, is either small (with Closed shrublands, Dense Evergreen Needleleaf Forests, and Marshland wetlands) or null in the remaining Wetlands classes and Forests classes (except Dense Broadleaf classes). This implies that, within 500 m pixels, the LULC products are less consistent within these classes, and it may exist remaining noise in one class from other classes. Since our dataset provides, as a metadata, the agreement percentage at pixel level, the user can always select the desired agreement threshold.

The collected time series data, in each LULC class, still contain some missing data that could be handled neither with the monthly aggregation process nor with the Terra-Aqua merging process (Table 10). For some classes, the average temporal availability percentage is very high (e.g., Grasslands, Shrublands, and Open Deciduous Broadleaf Forests). However, it is low for other classes (e.g., Moss and Lichen Lands, Marshland Wetlands, Marine Water Bodies, and Permanent Snow) which implies that their multi-spectral time series information is hugely affected by atmospheric and/or land conditions. For all LULC classes, it is noticeable that the average temporal availability percentage in Band 6 is low compared to the other bands which make Band 6 the most contaminated by gaps. The reason behind is the "dead lines" in Aqua band 6 caused by the already reported malfunctioning or noise in some of its detectors (Zhang et al., 2018b).

The unbalanced dataset (Table 9), and the balanced dataset corresponding to the 1000 pixels per class (Table 10), are distributed all over the world's GAUL partitions: ADM0 (i.e., countries) and ADM1 (e.g., departments, states, provinces). Each LULC class, in the two datasets, covers more than 6 countries and more than 13 departments, except Moss and Lichen Lands,

and Deciduous Needleleaf Forests that cover less number of countries and departments because of their natural scarce distribution over the world. Whereas, some of the LULC classes, namely Continental Water Bodies, Rainfed Croplands (Cereal and Broadleaf), and Urban and Built-Up Areas have a broad world coverage, i.e., more than 70 countries and more than 400 departments. In addition, the GHM index of the five cropland classes, and the Urban and Built-Up Areas class is widely higher (more than 59% of human change) compared to the other land cover classes, which proves their accurate annotation as human Land Uses. In a cultivated landscape, some plots may be in rotative fallow while other plots are being cultivated. Even though, the main signal from this class would come from the cultivated land because it is the main land use of the pixel.

**Table 8.** Description of the data quality control results of the 2900 pixels for each LULC level (L0 to L5) using the F1 score. The correspondence between long and short names of the LULC-classes is provided in Table 3

| L0 | F1 | L1 | F1 | L2 | F1 | L3 | F1 | L4 | F1 | L5 | F1 |
|---|---|---|---|---|---|---|---|---|---|---|---|
| Land Cover | 0.98 | Terrestrial Lands | 0.98 | BarrenLands | 0.88 | BarrenLands | 0.88 | BarrenLands | 0.88 | BarrenLands | 0.88 |
| | | | | MossAndLichen | NA | MossAndLichen | NA | MossAndLichen | NA | MossAndLichen | NA |
| | | | | Grasslands | 0.68 | Grasslands | 0.68 | Grasslands | 0.68 | Grasslands | 0.68 |
| | | | | Shrubland | 0.87 | ShrublandOpen | 0.74 | ShrublandOpen | 0.74 | ShrublandOpen | 0.74 |
| | | | | | | ShrublandClosed | 0.96 | ShrublandClosed | 0.96 | ShrublandClosed | 0.96 |
| | | | | Forests | 0.98 | ForestsDe | 0.99 | ForestsDeBr | 0.98 | ForestsOpDeBr | 0.81 |
| | | | | | | | | | | ForestsClDeBr | 0.79 |
| | | | | | | | | | | ForestsDeDeBr | 0.93 |
| | | | | | | | | ForestsDeNe | 1 | ForestsOpDeNe | 0.90 |
| | | | | | | | | | | ForestsClDeNe | 0.86 |
| | | | | | | | | | | ForestsDeDeNe | 0.91 |
| | | | | | | ForestsEv | 0.97 | ForestsEvBr | 0.94 | ForestsOpEvBr | 0.67 |
| | | | | | | | | | | ForestsClEvBr | 0.78 |
| | | | | | | | | | | ForestsDeEvBr | 0.93 |
| | | | | | | | | ForestsEvNe | 1 | ForestsOpEvNe | 0.85 |
| | | | | | | | | | | ForestsClEvNe | 0.84 |
| | | | | | | | | | | ForestsDeEvNe | 0.95 |
| | | | | PermanentSnow | 0.98 | PermanentSnow | 0.98 | PermanentSnow | 0.98 | PermanentSnow | 0.98 |
| | | Aquatic Lands | 0.94 | Wetland | 0.92 | WetlandMangro | 0.88 | WetlandMangro | 0.88 | WetlandMangro | 0.88 |
| | | | | | | WetlandSwamps | 0.91 | WetlandSwamps | 0.91 | WetlandSwamps | 0.91 |
| | | | | | | WetlandMarshl | 0.95 | WetlandMarshl | 0.95 | WetlandMarshl | 0.95 |
| | | | | WaterBody | 0.97 | WaterBodyMari | 0.88 | WaterBodyMari | 0.88 | WaterBodyMari | 0.88 |
| | | | | | | WaterBodyCont | 0.86 | WaterBodyCont | 0.86 | WaterBodyCont | 0.86 |
| Land Use | 0.93 | Crop Lands | 0.95 | CropSeasWater | 0.85 | CropSeasWater | 0.85 | CropSeasWater | 0.85 | CropSeasWater | 0.85 |
| | | | | CropCerea | 0.92 | CropCereaIrri | 0.92 | CropCereaIrri | 0.92 | CropCereaIrri | 0.92 |
| | | | | | | CropCereaRain | 0.91 | CropCereaRain | 0.91 | CropCereaRain | 0.91 |
| | | | | CropBroad | 0.98 | CropBroadIrri | 0.96 | CropBroadIrri | 0.96 | CropBroadIrri | 0.96 |
| | | | | | | CropBroadRain | 0.96 | CropBroadRain | 0.96 | CropBroadRain | 0.96 |
| | | UrbanBlUpArea | 0.93 | UrbanBlUpArea | 0.93 | UrbanBlUpArea | 0.93 | UrbanBlUpArea | 0.93 | UrbanBlUpArea | 0.93 |
| Mean | 0.96 | Mean | 0.95 | Mean | 0.91 | Mean | 0.90 | Mean | 0.91 | Mean | 0.87 |

**Table 9.** Description of the number of Food and Agricultural Organization's Global Administrative Unit Layers 2015 (GAUL) partitions ADM0 and ADM1, the average GHM index, and the average agreement percentage for the imbalanced version of the data. The number of ADM0 and ADM1 partitions for the class Water Bodies Marine (C21) is not provided because the GAUL partitions do not cover the seas and oceans.

| Class Id | # GAUL ADM0 | # GAUL ADM1 | Average GHM Index | Average Agreement Percentage |
|----------|-------------|-------------|-------------------|------------------------------|
| C1 | 29 | 207 | 16.46 | 100 |
| C2 | 2 | 6 | 0.018 | 100 |
| C3 | 42 | 248 | 18.09 | 100 |
| C4 | 26 | 108 | 4.37 | 100 |
| C5 | 11 | 28 | 8.43 | 96.84 |
| C6 | 19 | 72 | 13.80 | 85.80 |
| C7 | 20 | 97 | 13.03 | 86.16 |
| C8 | 15 | 53 | 18.47 | 100 |
| C9 | 1 | 8 | 2.65 | 81.32 |
| C10 | 3 | 9 | 3.31 | 85.51 |
| C11 | 3 | 14 | 2.90 | 96.09 |
| C12 | 12 | 26 | 4.12 | 81.72 |
| C13 | 36 | 92 | 12.82 | 86.57 |
| C14 | 39 | 188 | 3.88 | 100 |
| C15 | 6 | 23 | 0.47 | 80.83 |
| C16 | 16 | 87 | 5.57 | 86.14 |
| C17 | 65 | 460 | 10.04 | 97.35 |
| C18 | 29 | 50 | 13.94 | 87.65 |
| C19 | 37 | 87 | 12.96 | 86.95 |
| C20 | 9 | 14 | 9.58 | 97.16 |
| C21 | - | - | 0.19 | 100 |
| C22 | 121 | 916 | 1.15 | 100 |
| C23 | 187 | 209 | 0.30 | 100 |
| C24 | 30 | 146 | 59.72 | 100 |
| C25 | 45 | 204 | 75.09 | 100 |
| C26 | 101 | 954 | 61.14 | 100 |
| C27 | 23 | 141 | 72.82 | 100 |
| C28 | 83 | 593 | 60.88 | 100 |
| C29 | 185 | 1277 | 89.75 | 100 |

**Table 10:** Description of the number of GAUL partitions ADM0 and ADM1, the average GHM index, the average agreement percentage, and the average temporal availability percentage of each spectral band (B1 to B7) for the balanced version of the data. The number of ADM0 and ADM1 partitions for the class Water Bodies Marine (C21) is not provided because the GAUL partitions do not cover the seas and oceans.

| Class Id | # GAUL ADM0 | # GAUL ADM1 | Average GHM Index | Average Agreement Percentage | Average Temporal Availability Percentage | | | | | | |
|---|---|---|---|---|---|---|---|---|---|---|---|
| | | | | | B1 | B2 | B3 | B4 | B5 | B6 | B7 |
| C1 | 28 | 181 | 18.01 | 100 | 96.55 | 96.55 | 96.55 | 96.55 | 96.55 | 95.46 | 96.55 |
| C2 | 2 | 5 | 0.008 | 100 | 66.77 | 66.77 | 66.77 | 66.77 | 66.77 | 62.54 | 66.77 |
| C3 | 42 | 211 | 15.62 | 100 | 97.93 | 97.93 | 97.93 | 97.93 | 97.93 | 97.07 | 97.93 |
| C4 | 25 | 96 | 6.22 | 100 | 99.61 | 99.61 | 99.61 | 99.61 | 99.61 | 99.47 | 99.61 |
| C5 | 11 | 27 | 8.46 | 96.48 | 98.81 | 98.81 | 98.81 | 98.81 | 98.81 | 98.16 | 98.81 |
| C6 | 17 | 64 | 14.40 | 85.69 | 97.63 | 97.63 | 97.63 | 97.63 | 97.63 | 96.53 | 97.63 |
| C7 | 20 | 92 | 13.90 | 85.83 | 96.18 | 96.18 | 96.18 | 96.18 | 96.18 | 94.46 | 96.18 |
| C8 | 14 | 43 | 18.84 | 100 | 92.10 | 92.10 | 92.10 | 92.10 | 92.10 | 88.12 | 92.10 |
| C9 | 1 | 8 | 2.35 | 81.10 | 89.06 | 89.06 | 89.06 | 89.06 | 89.06 | 87.83 | 89.06 |
| C10 | 2 | 6 | 3.17 | 85.52 | 79.18 | 79.18 | 79.18 | 79.18 | 79.18 | 77.14 | 79.18 |
| C11 | 3 | 13 | 2.78 | 95.77 | 86.40 | 86.40 | 86.40 | 86.40 | 86.40 | 83.84 | 86.40 |
| C12 | 11 | 18 | 12.74 | 81.52 | 89.37 | 89.37 | 89.37 | 89.37 | 89.37 | 86.00 | 89.37 |
| C13 | 26 | 64 | 14.76 | 86.17 | 93.89 | 93.89 | 93.89 | 93.89 | 93.89 | 91.97 | 93.89 |
| C14 | 38 | 161 | 8.22 | 100 | 87.06 | 87.06 | 87.06 | 87.06 | 87.06 | 82.09 | 87.06 |
| C15 | 6 | 22 | 0.88 | 80.82 | 84.10 | 84.10 | 84.10 | 84.10 | 84.10 | 81.67 | 84.10 |
| C16 | 15 | 70 | 6.84 | 85.93 | 94.63 | 94.63 | 94.63 | 94.63 | 94.63 | 92.93 | 94.63 |
| C17 | 46 | 303 | 15.18 | 96.38 | 88.78 | 88.78 | 88.78 | 88.78 | 88.78 | 85.73 | 88.78 |
| C18 | 9 | 29 | 13.55 | 87.05 | 96.78 | 96.78 | 96.78 | 96.78 | 96.78 | 94.81 | 96.78 |
| C19 | 33 | 69 | 10.25 | 86.73 | 91.77 | 91.77 | 91.77 | 91.77 | 91.77 | 89.44 | 91.77 |
| C20 | 9 | 13 | 8.52 | 96.54 | 74.28 | 74.28 | 74.28 | 74.28 | 74.28 | 71.48 | 74.28 |
| C21 | - | - | 0.51 | 100 | 71.51 | 71.51 | 71.51 | 71.51 | 71.51 | 62.02 | 71.51 |
| C22 | 110 | 525 | 9.18 | 100 | 91.07 | 91.07 | 91.07 | 91.07 | 91.07 | 86.84 | 91.07 |
| C23 | 16 | 37 | 0.31 | 100 | 71.06 | 71.06 | 71.06 | 71.06 | 71.06 | 69.42 | 71.06 |
| C24 | 27 | 132 | 64.40 | 100 | 85.66 | 85.66 | 85.66 | 85.66 | 85.66 | 82.27 | 85.66 |
| C25 | 25 | 154 | 69.69 | 100 | 88.44 | 88.44 | 88.44 | 88.44 | 88.44 | 86.33 | 88.44 |
| C26 | 86 | 521 | 59.19 | 100 | 94.34 | 94.34 | 94.34 | 94.34 | 94.34 | 92.65 | 94.34 |

| Class Id | # GAUL ADM0 | # GAUL ADM1 | Average GHM Index | Average Agreement Percentage | Average Temporal Availability Percentage | | | | | | |
|---|---|---|---|---|---|---|---|---|---|---|---|
| | | | | | B1 | B2 | B3 | B4 | B5 | B6 | B7 |
| C27 | 23 | 123 | 69.36 | 100 | 91.13 | 91.13 | 91.13 | 91.13 | 91.13 | 89.31 | 91.13 |
| C28 | 77 | 428 | 60.69 | 100 | 94.66 | 94.66 | 94.66 | 94.66 | 94.66 | 93.13 | 94.66 |
| C29 | 144 | 655 | 83.19 | 100 | 85.61 | 85.61 | 85.61 | 85.61 | 85.61 | 82.50 | 85.61 |

## 5 Advantages, limitations, and potential applications of the dataset

The produced dataset is of high quality both in terms of the annotation and the generation of spectral reflectance. On the one hand, our dataset was annotated using the process of spatial-temporal combination of 15 global LULC products available in GEE. On the other hand, the time series of spectral reflectance were generated with less noise thanks to (1) the application of the quality assessment filters (MODLAND QA and State QA) in both MODIS products (MOD09A1 and MYD09A1), (2) the temporal aggregation from 8-day to monthly data, and (3) the Terra+Aqua merging process.

In addition, the annotation accuracy was assessed in two ways. First, thanks to the spatial-temporal agreement across the global GEE products, the level of consensus offered a cross-validation across independent products. Second, using a geographically representative sample of 2900 pixels (100 pixels per class selected by Algorithm 4) manually inspected by experts (visually photo-interpreted) using very high resolution imagery from both Google Earth and Bing Maps. Then, jointly agreed on which class each pixel corresponded to (agreement across interpreters according to Muchoney et al. (1999)). Thus the high quality of this dataset will certainly ensure the building of highly accurate machine learning models because building good quality machine learning models is possible only when trained on good quality data (García-Gil et al., 2019).

This smartly, pre-processed, and annotated dataset is targeted towards scientific users interested in developing and evaluating various DL models to detect LULC classes. For example, TimeSpec4LULC can be used i) to study the intra-class behaviors of LULCs, i.e., assess the behavior of one specific LULC in different areas of the world and see whether it maintains the same pattern or it reveals different patterns, and ii) to study the inter-class differences and similarities of LULCs, i.e., recognize and compare the patterns and dynamics of all LULCs (e.g., time series classification).

It appears that the dataset is only oriented towards LULC mapping since we provide for each time series a unique label. Nevertheless, this data can also be used i) to characterize the seasonal and inter-annual dynamics and changes of vegetation types and LULC classes, and ii) to perform environmental monitoring, management, and planning. This can be done by creating artificial dataset characterized by LULC change where each time series can have a sequence of annotations relative to each LULC type. Then, time series-based DL segmentation models can be trained on this artificial data and deployed on real time

series to detect and monitor the LULC change. On the other hand, the coordinates of each class provided in the dataset can also be used to detect anomalies in a specific class type, such as forest.

The dataset has also some limitations that we discuss below and to which we tried to provide some alternative solutions:

– Due to memory limitations in GEE, for some classes (Barren Lands, Grasslands, Dense Evergreen Broadleaf Forests, Water bodies, and Permanent snow), we could not export all the available pixels where the 15 LULC products agree. But
we are still providing representative data number (500,000 time series) distributed over the different GAUL partitions ADM0 and ADM1 where the class types exist.

– Even though we aggregated the original 8-day Terra and Aqua data into monthly composites, and merged both satellite monthly data into Terra+Aqua combined time series, we still have time series contaminated by missing values (Table 10). To overcome this issue and impute the time series, the user can apply different models, namely Recurrent Neural
Network (RNN) based time series imputation models, such as the Bidirectional Recurrent Imputation for Time Series (BRITS) (Cao et al., 2018), or the Generative Adversarial Network (GAN) based time series imputation models, for example the End-to-End Generative Adversarial Network (E2GAN) (Luo et al., 2019).

– For some classes, the agreement percentage is less than 100% (Wetlands classes, Closed Shrublands, and almost all Forests classes) because the 15 LULC products do not totally agree. In any case, we tried to slightly decrease the
400 agreement threshold and retain at least 1000 sample with the highest agreement percentage. In addition, we are providing at the pixel level the value of the agreement percentage so that the user can control the desired threshold and take it into consideration to evaluate the F1 score of the models.

– For some pixels, the ADM0-CODE and the ADM1-CODE are null because they are not provided by the GAUL product, especially for almost all the pixels of the class Water Bodies Marine (Table 9 and Table 10). This is obvious since the
405 GAUL partitions do not cover the seas and oceans.

– The number of 100 validated pixels is relatively small with regard to some classes containing a high number of pixels. The choice of this number was due to the challenging technical feasibility of the validation process and the lack of control resources. However, the pixels of each class were randomly selected following the maximum distance criteria described in Algorithm 4 which make them spatially representative of each class (Fig.10).

– The original dataset is highly imbalanced since there is a high variation of the number of time series between the different 29 classes. This imbalance is due to three reasons: (1) The spatial distribution of the different classes over the world, e.g., Barren Lands class is more world-dominant than Moss and Lichen Lands. (2) The agreement between the 15 LULC products, e.g., with an agreement threshold equal 1, the 15 products are compliant on 223,062 pixels in Open shrublands, and only on 9 pixels in Closed shrublands. (3) The temporal stability of the 29 classes, e.g., the two Water Bodies classes
are more consistent in time than the three Wetlands classes. To train machine learning models it is recommended to balance the dataset by selecting pixels evenly distributed over the world, with high agreement percentage, and with high

temporal availability percentage. Thus, we also provided a balanced subset of the original dataset containing 1000 pixels in each class, such that they are evenly distributed and representative of the globe (Algorithm 4).

## 6   Conclusions

Accurate LULC mapping is highly relevant for many applications, including Earth system modeling, environmental monitoring, management and planning, or natural hazards assessment, among many others. However, there still exists a high level of disagreement across current global LULC products, particularly for some LULC classes. To address the challenge of improving LULC products, we have created a smart open-source global dataset of multi-spectral time series for 29 LULC classes containing almost 6 million pixels annotated by using the spatial-temporal agreement across 15 global LULC products avail-
able in GEE. The 29 LULC classes were hierarchically grouped into a legend with five levels. The monthly 7-band time series dataset was made by merging the two MODIS sensor data records, Terra and Aqua, at 500 m resolution, and expands 22 years from 2000 to the end of 2021. Each pixel is provided with a set of meta-data about geographic coordinates, country and departmental divisions, spatial-temporal consistency across LULC products, temporal data availability, and the global human modification index. Finally, to assess the annotation quality of the dataset, a sample of 100 pixels per class, evenly distributed
around the world, was selected by maximizing the distance among sampled pixels, and validated with photo-interpretation by experts using very high resolution images from both Google Earth and Bing Maps. The overall F1-score of the annotation varied from 96% at the coarser classification level to 87% at the finest level. This smartly, pre-processed, and annotated dataset is targeted towards scientific users interested in developing and evaluating various machine learning models, including deep learning networks, to perform global LULC mapping.

## 7   Code and data availability

This dataset (Version 1.2) (Khaldi et al., 2022) is available to the public through an unrestricted data repository hosted by Zenodo at:

https://zenodo.org/record/5913554#.Yfvwj-rMK70

*Author contributions.*   RK contributed to the conception of the dataset, implemented the code, performed all the data extraction and wrote
the paper. DA-S contributed to the conception of the dataset, assessed its quality, provided guidance, and wrote the paper. EG assessed the quality of the dataset. YB contributed to the conception of the dataset. AE and FH provided edits and suggestions. ST contributed to the conception of the dataset, provided guidance, and wrote the paper.

*Competing interests.*   The authors declare that they have no conflict of interest.

*Acknowledgements.* This work was partially supported by DETECTOR (A-RNM-256-UGR18 Universidad de Granada/FEDER), LifeWatch
SmartEcomountains (LifeWatch-2019-10-UGR-01 Ministerio de Ciencia e Innovación/Universidad de Granada/FEDER), BBVA DeepSCOP
(Ayudas Fundación BBVA a Equipos de Investigación Científica 2018), DeepL-ISCO (A-TIC-458-UGR18 Ministerio de Ciencia e Inno-
vación/FEDER), SMART-DASCI (TIN2017-89517-P Ministerio de Ciencia e Innovación/Universidad de Granada/FEDER), BigDDL-CET
(P18-FR-4961 Ministerio de Ciencia e Innovación/Universidad de Granada/FEDER), RESISTE (P18-RT-1927 Consejería de Economía,
Conocimiento, y Universidad from the Junta de Andalucía/FEDER), and Ecopotential (641762 European Commission).

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

# Appendix A

**Table A1.** Translation of TimeSpec4LULC legend to FAO's Land Cover Classification System (LCCS).

| FAO's LCCS System | | | TimeSpec4LULC nomenclature |
|---|---|---|---|
| Dichotomous phase 1 | Dichotomous phase 2 | Dichotomous phase 3 | |
| A. Primarily vegetated | A1. Terrestrial | A11.Cultivated and Managed Terrestrial Areas. | C25 Irrigated cereal croplands |
| | | | C26 Rainfed cereal croplands |
| | | | C27 Irrigated broadleaf croplands |
| | | | C28 Rainfed broadleaf croplands |
| | | A12.Natural and Semi-Natural Terrestrial Vegetation. | C2 Moss and lichen lands |
| | | | C3 Grasslands |
| | | | C4 Open shrublands |
| | | | C5 Closed shrublands |
| | | | C6 Open Deciduous Broadleaf Forests |
| | | | C7 Closed Deciduous Broadleaf Forests |
| | | | C8 Dense Deciduous Broadleaf Forests |
| | | | C9 Open Deciduous Needleleaf Forests |
| | | | C10 Closed Deciduous Needleleaf Forests |
| | | | C11 Dense Deciduous Needleleaf Forests |
| | | | C12 Open Evergreen Broadleaf Forests |
| | | | C13 Closed Evergreen Broadleaf Forests |
| | | | C14 Dense Evergreen Broadleaf Forests |
| | | | C15 Open Evergreen Needleleaf Forests |
| | | | C16 Closed Evergreen Needleleaf Forests |
| | | | C17 Dense Evergreen Needleleaf Forests |
| | A2. Aquatic or regularly flooded | A23.Cultivated Aquatic or Regularly Flooded Areas. | C24 Croplands flooded with seasonal water |
| | | A24.Natural and Semi-Natural Aquatic or Regularly Flooded Vegetation. | C18 Mangrove wetlands |
| | | | C19 Swamp wetlands |
| | | | C20 Marshland wetlands |
| B. Primarily non-vegetated | B1. Terrestrial | B15.Artificial Surfaces and Associated Areas. | |
| | | B16.Bare Areas. | C1 Barren Lands |
| | B2. Aquatic or regularly flooded | B27.Artificial Water Bodies, Snow and Ice. | |
| | | B28.Natural Water Bodies, Snow and Ice. | C21 Marine water bodies |
| | | | C22 Continental water bodies |
| | | | C23 Permanent snow |

**Table A2:** Detailed description of each LULC class used to build the legend for TimeSpec4LULC dataset. NU: Not Used. NA: Not Available.

| Class | Product | Description |
|---|---|---|
| **C1** | P1 | 16. Barren: at least 60% of area is non-vegetated barren (sand, rock, soil) areas with less than 10% vegetation. |
| | P2 | 15. Non-Vegetated Lands: at least 60% of area is non-vegetated barren (sand, rock, soil) or permanent snow and ice with less than 10% vegetation. |
| | P3 | NA |
| | P4 | 7. Non-Vegetated Lands: at least 60% of area is non-vegetated barren (sand, rock, soil) or permanent snow/ice with less than 10% vegetation. |
| | P5 | 11. Non-Vegetated Lands: at least 60% of area is non-vegetated barren (sand, rock, soil) with less than 10% vegetation. |
| | P6 | 60. Bare / sparse vegetation. Lands with exposed soil, sand, or rocks and never has more than 10 % vegetated cover during any time of the year. |
| | P7 | Tree canopy cover<10% |
| | P8 | 200. Bare areas |
| | P9 | 0. Non cropland |
| | P10 | 2. Non forest |
| | P11 | (Tree cover <10%) AND (gain=0) AND (loss=0) AND (datamask $\neq$ 2. Permanent water bodies) |
| | P12 | Tree heights<1m |
| | P13 | (1. Not water) OR (0. No data) |
| | P14 | 0. max_extent |
| | P15 | Not($\geq 1$) |
| **C2** | P1 | 16. Barren: at least 60% of area is non-vegetated barren (sand, rock, soil) areas with less than 10% vegetation. |
| | P2 | 15. Non-Vegetated Lands: at least 60% of area is non-vegetated barren (sand, rock, soil) or permanent snow and ice with less than 10% vegetation. |
| | P3 | NA |
| | P4 | 7. Non-Vegetated Lands: at least 60% of area is non-vegetated barren (sand, rock, soil) or permanent snow/ice with less than 10% vegetation. |
| | P5 | 11. Non-Vegetated Lands: at least 60% of area is non-vegetated barren (sand, rock, soil) with less than 10% vegetation. |
| | P6 | 100. Moss and lichen |
| | P7 | Tree canopy cover<10% |
| | P8 | (200. Bare areas) OR (150. Sparse (>15%) vegetation (woody vegetation, shrubs, grassland)) |
| | P9 | 0. Non cropland |
| | P10 | 2. Non forest |
| | P11 | (Tree cover <10%) AND (gain=0) AND (loss=0) AND (datamask $\neq$ 2. Permanent water bodies) |
| | P12 | Tree heights<1m |

| | | |
|---|---|---|
| | P13 | (1. Not water) OR (0. No data) |
| | P14 | 0. max_extent |
| | P15 | Not($\geq 1$) |
| **C3** | P1 | 10. Grasslands: dominated by herbaceous annuals (<2m). |
| | P2 | 10. Grasslands: dominated by herbaceous annuals (<2m). |
| | P3 | 1. Grasslands: dominated by herbaceous annuals (<2m) including cereal croplands. |
| | P4 | 6. Annual Grass Vegetation: dominated by herbaceous annuals (<2m) including cereal croplands. |
| | P5 | 6. Grass: dominated by herbaceous annuals (<2m) that are not cultivated. |
| | P6 | 30.Herbaceous vegetation. Plants without persistent stem or shoots above ground and lacking definite firm structure. Tree and shrub cover is less than 10 %. |
| | P7 | Tree canopy cover<10% |
| | P8 | 140. Closed to open (>15%) grassland |
| | P9 | NA |
| | P10 | 2. Non forest |
| | P11 | (Tree cover <10%) AND (gain=0) AND (loss=0) AND (datamask $\neq$ 2. Permanent water bodies) |
| | P12 | Tree heights<2m |
| | P13 | (1. Not water) OR (0. No data) |
| | P14 | 0. max_extent |
| | P15 | Not($\geq 1$) |
| **C4** | P1 | 7. Open Shrublands: dominated by woody perennials (1-2m height) 10-60% cover. |
| | P2 | 7. Open Shrublands: dominated by woody perennials (1-2m height) 10-60% cover. |
| | P3 | 2. Shrublands: shrub (1-2m) cover >10%. |
| | P4 | NA |
| | P5 | 5. Shrub: Shrub (1-2m) cover >10%. |
| | P6 | (20. Shrubs. Woody perennial plants with persistent and woody stems and without any defined main stem being less than 5 m tall. The shrub foliage can be either evergreen or deciduous.) AND (10 <shrub-coverfraction <50) |
| | P7 | Tree canopy cover<10% |
| | P8 | 150. Sparse (>15%) vegetation (woody vegetation, shrubs, grassland) |
| | P9 | 0. Non cropland |
| | P10 | 2. Non forest |
| | P11 | (Tree cover <10%) AND (gain=0) AND (loss=0) AND (datamask $\neq$ 2. Permanent water bodies) |
| | P12 | Tree heights<2m |
| | P13 | (1. Not water) OR (0. No data) |
| | P14 | 0. max_extent |
| | P15 | Not($\geq 1$) |

| | | |
|---|---|---|
| | P1 | 6. Closed Shrublands: dominated by woody perennials (1-2m height) >60% cover. |
| | P2 | 6. Closed Shrublands: dominated by woody perennials (1-2m height) >60% cover. |
| | P3 | 2. Shrublands: shrub (1-2m) cover >10%. |
| | P4 | NA |
| | P5 | 5. Shrub: Shrub (1-2m) cover >10%. |
| | P6 | (20. Shrubs. Woody perennial plants with persistent and woody stems and without any defined main stem being less than 5 m tall. The shrub foliage can be either evergreen or deciduous.) AND (shrub-coverfraction >50) |
| **C5** | P7 | Tree canopy cover<10% |
| | P8 | 130. Closed to open (>15%) shrubland (<5m) |
| | P9 | 0. Non cropland |
| | P10 | 2. Non forest |
| | P11 | (Tree cover <10%) AND (gain=0) AND (loss=0) AND (datamask $\neq$ 2. Permanent water bodies) |
| | P12 | Tree heights<2m |
| | P13 | (1. Not water) OR (0. No data) |
| | P14 | 0. max_extent |
| | P15 | Not($\geq 1$) |
| | P1 | NA |
| | P2 | NA |
| | P3 | NA |
| | P4 | 4. Deciduous Broadleaf Vegetation: dominated by deciduous broadleaf trees and shrubs (>1m). Woody vegetation cover >10%. |
| | P5 | 4. Deciduous Broadleaf Trees: dominated by deciduous broadleaf trees (>2m). Tree cover >10%. |
| | P6 | (15% <tree-coverfraction <30%) ADD (4. Deciduous broad leaf) |
| **C6** | P7 | 15%<Tree canopy cover<30% |
| | P8 | 60. Open (15-40%) broadleaved deciduous forest (>5m) |
| | P9 | NA |
| | P10 | 1. Forest |
| | P11 | (15<Tree cover <30%) AND (gain=0) AND (loss=0) AND (datamask $\neq$ 2. Permanent water bodies) |
| | P12 | Tree heights>2m |
| | P13 | (1. Not water) OR (0. No data) |
| | P14 | 0. max_extent |
| | P15 | Not($\geq 1$) |

| | | | |
|---|---|---|---|
| **C7** | P1 | NA | |
| | P2 | NA | |
| | P3 | NA | |
| | P4 | 4. Deciduous Broadleaf Vegetation: dominated by deciduous broadleaf trees and shrubs (>1m). Woody vegetation cover >10%. | |
| | P5 | 4. Deciduous Broadleaf Trees: dominated by deciduous broadleaf trees (>2m). Tree cover >10%. | |
| | P6 | (40% <tree-coverfraction <60%) ADD (4. Deciduous broad leaf) | |
| | P7 | 40% <Tree canopy cover <60% | |
| | P8 | 50. Closed (>40%) broadleaved deciduous forest (>5m) | |
| | P9 | NA | |
| | P10 | 1. Forest | |
| | P11 | (40 <Tree cover <60%) AND (gain=0) AND (loss=0) AND (datamask $\neq$ 2. Permanent water bodies) | |
| | P12 | Tree heights >2m | |
| | P13 | (1. Not water) OR (0. No data) | |
| | P14 | 0. max_extent | |
| | P15 | Not($\geq$ 1) | |
| **C8** | P1 | 4. Deciduous Broadleaf Forests: dominated by deciduous broadleaf trees (canopy >2m). Tree cover >60%. | |
| | P2 | 4. Deciduous Broadleaf Forests: dominated by deciduous broadleaf trees (canopy >2m). Tree cover >60%. | |
| | P3 | 6. Deciduous Broadleaf Forests: dominated by deciduous broadleaf trees (canopy >2m). Tree cover >60%. | |
| | P4 | 4. Deciduous Broadleaf Vegetation: dominated by deciduous broadleaf trees and shrubs (>1m). Woody vegetation cover >10%. | |
| | P5 | 4. Deciduous Broadleaf Trees: dominated by deciduous broadleaf trees (>2m). Tree cover >10%. | |
| | P6 | (tree-coverfraction >60%) ADD (4. Deciduous broad leaf) | |
| | P7 | Tree canopy cover>60% | |
| | P8 | 50. Closed (>40%) broadleaved deciduous forest (>5m). | |
| | P9 | NA | |
| | P10 | 1. Forest | |
| | P11 | (Tree cover >60%) AND (gain=0) AND (loss=0) AND (datamask $\neq$ 2. Permanent water bodies) | |
| | P12 | Tree heights>2m | |
| | P13 | (1. Not water) OR (0. No data) | |
| | P14 | 0. max_extent | |
| | P15 | Not($\geq$ 1) | |

| | | | |
|---|---|---|---|
| **C9** | P1 | NA | |
| | P2 | NA | |
| | P3 | NA | |
| | P4 | 2. Evergreen Broadleaf Vegetation: dominated by evergreen broadleaf and palmate trees and shrubs (>1m). Woody vegetation cover >10%. | |
| | P5 | 2. Evergreen Broadleaf Trees: dominated by evergreen broadleaf and palmate trees (>2m). Tree cover >10%. | |
| | P6 | (15% <tree-coverfraction <30%) ADD (2. Evergreen broad leaf) | |
| | P7 | 15%<Tree canopy cover<30% | |
| | P8 | 40. Closed to open (>15%) broadleaved evergreen and/or semi-deciduous forest (>5m) | |
| | P9 | NA | |
| | P10 | 1. Forest | |
| | P11 | (15<Tree cover <30%) AND (gain=0) AND (loss=0) AND (datamask $\neq$ 2. Permanent water bodies) | |
| | P12 | Tree heights>2m | |
| | P13 | (1. Not water) OR (0. No data) | |
| | P14 | 0. max_extent | |
| | P15 | Not($\geq$ 1) | |
| **C10** | P1 | NA | |
| | P2 | NA | |
| | P3 | NA | |
| | P4 | 2. Evergreen Broadleaf Vegetation: dominated by evergreen broadleaf and palmate trees and shrubs (>1m). Woody vegetation cover >10%. | |
| | P5 | 2. Evergreen Broadleaf Trees: dominated by evergreen broadleaf and palmate trees (>2m). Tree cover >10%. | |
| | P6 | (40% <tree-coverfraction <60%) ADD (2. Evergreen broad leaf) | |
| | P7 | 40%<Tree canopy cover<60% | |
| | P8 | 40. Closed to open (>15%) broadleaved evergreen and/or semi-deciduous forest (>5m) | |
| | P9 | NA | |
| | P10 | 1. Forest | |
| | P11 | (40<Tree cover <60%) AND (gain=0) AND (loss=0) AND (datamask $\neq$ 2. Permanent water bodies) | |
| | P12 | Tree heights>2m | |
| | P13 | (1. Not water) OR (0. No data) | |
| | P14 | 0. max_extent | |
| | P15 | Not($\geq$ 1) | |

| | | |
|---|---|---|
| **C11** | P1 | 2. Evergreen Broadleaf Forests: dominated by evergreen broadleaf and palmate trees (canopy >2m). Tree cover >60%. |
| | P2 | 2. Evergreen Broadleaf Forests: dominated by evergreen broadleaf and palmate trees (canopy >2m). Tree cover >60%. |
| | P3 | 5. Evergreen Broadleaf Forests: dominated by evergreen broadleaf and palmate trees (canopy >2m). Tree cover >60%. |
| | P4 | 2. Evergreen Broadleaf Vegetation: dominated by evergreen broadleaf and palmate trees and shrubs (>1m). Woody vegetation cover >10%. |
| | P5 | 2. Evergreen Broadleaf Trees: dominated by evergreen broadleaf and palmate trees (>2m). Tree cover >10%. |
| | P6 | (tree-coverfraction >60%) ADD (2. Evergreen broad leaf) |
| | P7 | Tree canopy cover>60% |
| | P8 | 40. Closed to open (>15%) broadleaved evergreen and/or semi-deciduous forest (>5m) |
| | P9 | NA |
| | P10 | 1. Forest |
| | P11 | (Tree cover >60%) AND (gain=0) AND (loss=0) AND (datamask $\neq$ 2. Permanent water bodies) |
| | P12 | Tree heights>2m |
| | P13 | (1. Not water) OR (0. No data) |
| | P14 | 0. max_extent |
| | P15 | Not($\geq 1$) |
| **C12** | P1 | NA |
| | P2 | NA |
| | P3 | NA |
| | P4 | 3. Deciduous Needleleaf Vegetation: dominated by deciduous needleleaf (larch) trees and shrubs (>1m). Woody vegetation cover >10%. |
| | P5 | 3. Deciduous Needleleaf Trees: dominated by deciduous needleleaf (larch) trees (>2m). Tree cover >10%. |
| | P6 | (15<tree-coverfraction <30%) ADD (3. Deciduous needle leaf) |
| | P7 | 15%<Tree canopy cover<30% |
| | P8 | NA |
| | P9 | NA |
| | P10 | 1. Forest |
| | P11 | (15<Tree cover <30%) AND (gain=0) AND (loss=0) AND (datamask $\neq$ 2. Permanent water bodies) |
| | P12 | Tree heights>2m |
| | P13 | (1. Not water) OR (0. No data) |
| | P14 | 0. max_extent |
| | P15 | Not($\geq 1$) |

| | | |
|---|---|---|
| | P1 | NA |
| | P2 | NA |
| | P3 | NA |
| | P4 | 3. Deciduous Needleleaf Vegetation: dominated by deciduous needleleaf (larch) trees and shrubs (>1m). Woody vegetation cover >10%. |
| | P5 | 3. Deciduous Needleleaf Trees: dominated by deciduous needleleaf (larch) trees (>2m). Tree cover >10%. |
| | P6 | (40<tree-coverfraction <60%) ADD (3. Deciduous needle leaf) |
| **C13** | P7 | 40%<Tree canopy cover<60% |
| | P8 | NA |
| | P9 | NA |
| | P10 | 1. Forest |
| | P11 | (40<Tree cover <60%) AND (gain=0) AND (loss=0) AND (datamask $\neq$ 2. Permanent water bodies) |
| | P12 | Tree heights>2m |
| | P13 | (1. Not water) OR (0. No data) |
| | P14 | 0. max_extent |
| | P15 | Not($\geq$ 1) |
| | P1 | 3. Deciduous Needleleaf Forests: dominated by deciduous needleleaf (larch) trees (canopy >2m). Tree cover >60%. |
| | P2 | 3. Deciduous Needleleaf Forests: dominated by deciduous needleleaf (larch) trees (canopy >2m). Tree cover >60%. |
| | P3 | 8. Deciduous Needleleaf Forests: dominated by deciduous needleleaf (larch) trees (canopy >2m). Tree cover >60%. |
| | P4 | 3. Deciduous Needleleaf Vegetation: dominated by deciduous needleleaf (larch) trees and shrubs (>1m). Woody vegetation cover >10%. |
| | P5 | 3. Deciduous Needleleaf Trees: dominated by deciduous needleleaf (larch) trees (>2m). Tree cover >10%. |
| | P6 | (tree-coverfraction >60%) ADD (3. Deciduous needle leaf) |
| **C14** | P7 | Tree canopy cover>60% |
| | P8 | NA |
| | P9 | NA |
| | P10 | 1. Forest |
| | P11 | (Tree cover >60%) AND (gain=0) AND (loss=0) AND (datamask $\neq$ 2. Permanent water bodies) |
| | P12 | Tree heights>2m |
| | P13 | (1. Not water) OR (0. No data) |
| | P14 | 0. max_extent |
| | P15 | Not($\geq$ 1) |

| | | |
|---|---|---|
| **C15** | P1 | 9. Savannas: tree cover 10-30% (canopy >2m). |
| | P2 | 9. Savannas: tree cover 10-30% (canopy >2m). |
| | P3 | NA |
| | P4 | 1. Evergreen Needleleaf Vegetation: dominated by evergreen conifer trees and shrubs (>1m). Woody vegetation cover >10%. |
| | P5 | 1. Evergreen Needleleaf Trees: dominated by evergreen conifer trees (>2m). Tree cover >10%. |
| | P6 | (15%<tree-coverfraction<30%) ADD (1. Evergreen needle leaf) |
| | P7 | 15%<Tree canopy cover<30% |
| | P8 | 90. Open (15-40%) needleleaved deciduous or evergreen forest (>5m) |
| | P9 | NA |
| | P10 | 1. Forest |
| | P11 | (15%<Tree cover<30%) AND (gain=0) AND (loss=0) AND (datamask $\neq$ 2. Permanent water bodies) |
| | P12 | Tree heights>2m |
| | P13 | (1. Not water) OR (0. No data) |
| | P14 | 0. max_extent |
| | P15 | Not($\geq$ 1) |
| **C16** | P1 | 8. Woody Savannas: tree cover 30-60% (canopy >2m). |
| | P2 | 8. Woody Savannas: tree cover 30-60% (canopy >2m). |
| | P3 | 4. Savannas: between 10-60% tree cover (>2m). |
| | P4 | 1. Evergreen Needleleaf Vegetation: dominated by evergreen conifer trees and shrubs (>1m). Woody vegetation cover >10%. |
| | P5 | 1. Evergreen Needleleaf Trees: dominated by evergreen conifer trees (>2m). Tree cover >10%. |
| | P6 | (40%<tree-coverfraction<60%) ADD (1. Evergreen needle leaf) |
| | P7 | 40%<Tree canopy cover<60% |
| | P8 | 70. Closed (>40%) needleleaved evergreen forest (>5m) |
| | P9 | NA |
| | P10 | 1. Forest |
| | P11 | (40%<Tree cover<60%) AND (gain=0) AND (loss=0) AND (datamask $\neq$ 2. Permanent water bodies) |
| | P12 | Tree heights>2m |
| | P13 | (1. Not water) OR (0. No data) |
| | P14 | 0. max_extent |
| | P15 | Not($\geq$ 1) |

| | | |
|---|---|---|
| **C17** | P1 | 1. Evergreen Needleleaf Forests: dominated by evergreen conifer trees (canopy >2m). Tree cover >60%. |
| | P2 | 1. Evergreen Needleleaf Forests: dominated by evergreen conifer trees (canopy >2m). Tree cover >60%. |
| | P3 | 7. Evergreen Needleleaf Forests: dominated by evergreen conifer trees (canopy >2m). Tree cover >60%. |
| | P4 | 1. Evergreen Needleleaf Vegetation: dominated by evergreen conifer trees and shrubs (>1m). Woody vegetation cover >10%. |
| | P5 | 1. Evergreen Needleleaf Trees: dominated by evergreen conifer trees (>2m). Tree cover >10%. |
| | P6 | (tree-coverfraction>60%) ADD (1. Evergreen needle leaf) |
| | P7 | Tree canopy cover>60% |
| | P8 | 70. Closed (>40%) needleleaved evergreen forest (>5m) |
| | P9 | NA |
| | P10 | 1. Forest |
| | P11 | (Tree cover>60%) AND (gain=0) AND (loss=0) AND (datamask $\neq$ 2. Permanent water bodies) |
| | P12 | Tree heights>2m |
| | P13 | (1. Not water) OR (0. No data) |
| | P14 | 0. max_extent |
| | P15 | Not($\geq$ 1) |
| **C18** | P1 | 11. Permanent Wetlands: permanently inundated lands with 30-60% water cover and >10% vegetated cover. |
| | P2 | 11. Permanent Wetlands: permanently inundated lands with 30-60% water cover and >10% vegetated cover. |
| | P3 | NA |
| | P4 | NA |
| | P5 | NA |
| | P6 | 90. Herbaceous wetland. Lands with a permanent mixture of water and herbaceous or woody vegetation. The vegetation can be present in either salt, brackish, or fresh water. |
| | P7 | Tree canopy cover>10% |
| | P8 | 170. Closed (>40%) broadleaved semi-deciduous and/or evergreen forest regularly flooded - saline water |
| | P9 | NA |
| | P10 | NA |
| | P11 | (Tree cover>10%) AND (gain=0) AND (loss=0) OR (datamask = 2. Permanent water bodies) |
| | P12 | Tree heights>2m |
| | P13 | (2. Seasonal water) OR (3. Permanent water) |
| | P14 | 1. max_extent |
| | P15 | Not($\geq$ 1) |

| | | |
|---|---|---|
| **C19** | P1 | 11. Permanent Wetlands: permanently inundated lands with 30-60% water cover and >10% vegetated cover. |
| | P2 | 11. Permanent Wetlands: permanently inundated lands with 30-60% water cover and >10% vegetated cover. |
| | P3 | NA |
| | P4 | NA |
| | P5 | NA |
| | P6 | 90. Herbaceous wetland. Lands with a permanent mixture of water and herbaceous or woody vegetation. The vegetation can be present in either salt, brackish, or fresh water. |
| | P7 | Tree canopy cover>10% |
| | P8 | (180. Closed to open (>15%) vegetation (grassland, shrubland, woody vegetation) on regularly flooded or waterlogged soil - fresh, brackish or saline water) OR (160. Closed (>40%) broadleaved forest regularly flooded - Fresh water) |
| | P9 | NA |
| | P10 | NA |
| | P11 | (Tree cover>10%) AND (gain=0) AND (loss=0) OR (datamask = 2. Permanent water bodies) |
| | P12 | Tree heights>2m |
| | P13 | (2. Seasonal water) OR (3. Permanent water) |
| | P14 | 1. max_extent |
| | P15 | Not($\geq$ 1) |
| **C20** | P1 | 11. Permanent Wetlands: permanently inundated lands with 30-60% water cover and >10% vegetated cover. |
| | P2 | 11. Permanent Wetlands: permanently inundated lands with 30-60% water cover and >10% vegetated cover. |
| | P3 | NA |
| | P4 | NA |
| | P5 | NA |
| | P6 | 90. Herbaceous wetland. Lands with a permanent mixture of water and herbaceous or woody vegetation. The vegetation can be present in either salt, brackish, or fresh water. |
| | P7 | Tree canopy cover<10% |
| | P8 | (180. Closed to open (>15%) vegetation (grassland, shrubland, woody vegetation) on regularly flooded or waterlogged soil - fresh, brackish or saline water) OR (170. Closed (>40%) broadleaved semi-deciduous and/or evergreen forest regularly flooded - saline water) OR (160. Closed (>40%) broadleaved forest regularly flooded - Fresh water) |
| | P9 | NA |
| | P10 | NA |
| | P11 | (Tree cover<10%) AND (gain=0) AND (loss=0) OR (datamask = 2. Permanent water bodies) |
| | P12 | Tree heights<2m |
| | P13 | (2. Seasonal water) OR (3. Permanent water) |
| | P14 | 1. max_extent |
| | P15 | Not($\geq$ 1) |

| | | |
|---|---|---|
| **C21** | P1 | 17. Water Bodies: at least 60% of area is covered by permanent water bodies. |
| | P2 | 0. Water Bodies: at least 60% of area is covered by permanent water bodies. |
| | P3 | 0. Water Bodies: at least 60% of area is covered by permanent water bodies. |
| | P4 | 0. Water Bodies: at least 60% of area is covered by permanent water bodies. |
| | P5 | 0. Water Bodies: at least 60% of area is covered by permanent water bodies |
| | P6 | 200. Oceans, seas. Can be either fresh or salt-water bodies. |
| | P7 | NA |
| | P8 | 210. Water bodies |
| | P9 | NA |
| | P10 | 3. Water |
| | P11 | NA |
| | P12 | NA |
| | P13 | 3. Permanent water |
| | P14 | 1. max_extent |
| | P15 | Not($\geq 1$) |
| **C22** | P1 | 17. Water Bodies: at least 60% of area is covered by permanent water bodies. |
| | P2 | 0. Water Bodies: at least 60% of area is covered by permanent water bodies. |
| | P3 | 0. Water Bodies: at least 60% of area is covered by permanent water bodies. |
| | P4 | 0. Water Bodies: at least 60% of area is covered by permanent water bodies. |
| | P5 | 0. Water Bodies: at least 60% of area is covered by permanent water bodies |
| | P6 | 80. Permanent water bodies. Lakes, reservoirs, and rivers. Can be either fresh or salt-water bodies. |
| | P7 | NA |
| | P8 | 210. Water bodies |
| | P9 | NA |
| | P10 | 3. Water |
| | P11 | NA |
| | P12 | NA |
| | P13 | 3. Permanent water |
| | P14 | 1. max_extent |
| | P15 | Not($\geq 1$) |

| | | |
|---|---|---|
| **C23** | P1 | 15.Permanent Snow and Ice: at least 60% of area is covered by snow and ice for at least 10 months of the year. |
| | P2 | NA |
| | P3 | NA |
| | P4 | NA |
| | P5 | 10. Permanent Snow and Ice: at least 60% of area is covered by snow and ice for at least 10 months of the year. |
| | P6 | 70. Snow and ice. Lands under snow or ice cover throughout the year. |
| | P7 | NA |
| | P8 | 220. Permanent snow and ice |
| | P9 | NA |
| | P10 | NA |
| | P11 | NA |
| | P12 | NA |
| | P13 | (1. Not water) OR (0. No data) |
| | P14 | 0. max_extent |
| | P15 | Not($\geq 1$) |
| **C24** | P1 | 12. Croplands: at least 60% of area is cultivated cropland. |
| | P2 | 12. Croplands: at least 60% of area is cultivated cropland. |
| | P3 | (3. Broadleaf Croplands: bominated by herbaceous annuals (<2m) that are cultivated with broadleaf crops.) OR (1. Grasslands: dominated by herbaceous annuals (<2m) including cereal croplands.) |
| | P4 | (5. Annual Broadleaf Vegetation: dominated by herbaceous annuals (<2m). At least 60% cultivated broadleaf crops.) OR (6. Annual Grass Vegetation: dominated by herbaceous annuals (<2m) including cereal croplands.) |
| | P5 | (8. Broadleaf Croplands: dominated by herbaceous annuals (<2m). At least 60% cultivated broadleaf crops.) OR (7. Cereal Croplands: dominated by herbaceous annuals (<2m). At least 60% cultivated cereal crops.) |
| | P6 | 40. Cultivated and managed vegetation / agriculture. Lands covered with temporary crops followed by harvest and a bare soil period (e.g., single and multiple cropping systems). Note that perennial woody crops will be classified as the appropriate forest or shrub land cover type. |
| | P7 | NA |
| | P8 | (11. Post-flooding or irrigated croplands) OR (14. Rainfed croplands) |
| | P9 | (1. Croplands: irrigation major) OR (2. Croplands: irrigation minor) OR (3. Croplands: rainfed) OR (4. Croplands: rainfed, minor fragments) OR (5. Croplands: rainfed, very minor fragments) |
| | P10 | NA |
| | P11 | NA |
| | P12 | NA |
| | P13 | (2. Seasonal water) OR (3. Permanent water) |
| | P14 | (0. No change) OR (4. Seasonal) OR (5. New seasonal) OR (8. Permanent to seasonal) OR (10. Ephemeral seasonal) |
| | P15 | Not($\geq 1$) |

| | | |
|---|---|---|
| **C25** | P1 | 12. Croplands: at least 60% of area is cultivated cropland. |
| | P2 | 12. Croplands: at least 60% of area is cultivated cropland. |
| | P3 | 1. Grasslands: dominated by herbaceous annuals (<2m) including cereal croplands. |
| | P4 | 6. Annual Grass Vegetation: dominated by herbaceous annuals (<2m) including cereal croplands. |
| | P5 | 7. Cereal Croplands: dominated by herbaceous annuals (<2m). At least 60% cultivated cereal crops. |
| | P6 | 40. Cultivated and managed vegetation / agriculture. Lands covered with temporary crops followed by harvest and a bare soil period (e.g., single and multiple cropping systems). Note that perennial woody crops will be classified as the appropriate forest or shrub land cover type. |
| | P7 | NA |
| | P8 | 11. Post-flooding or irrigated croplands |
| | P9 | (1. Croplands: irrigation major) OR (2. Croplands: irrigation minor) |
| | P10 | NA |
| | P11 | NA |
| | P12 | NA |
| | P13 | (1. Not water) OR (0. No data) |
| | P14 | 0. max_extent |
| | P15 | Not($\geq 1$) |
| **C26** | P1 | 12. Croplands: at least 60% of area is cultivated cropland. |
| | P2 | 12. Croplands: at least 60% of area is cultivated cropland. |
| | P3 | 1. Grasslands: dominated by herbaceous annuals (<2m) including cereal croplands. |
| | P4 | 6. Annual Grass Vegetation: dominated by herbaceous annuals (<2m) including cereal croplands. |
| | P5 | 7. Cereal Croplands: dominated by herbaceous annuals (<2m). At least 60% cultivated cereal crops. |
| | P6 | 40. Cultivated and managed vegetation / agriculture. Lands covered with temporary crops followed by harvest and a bare soil period (e.g., single and multiple cropping systems). Note that perennial woody crops will be classified as the appropriate forest or shrub land cover type. |
| | P7 | NA |
| | P8 | 14. Rainfed croplands |
| | P9 | (3. Croplands: rainfed) OR (4. Croplands: rainfed, minor fragments) OR (5. Croplands: rainfed, very minor fragments) |
| | P10 | NA |
| | P11 | NA |
| | P12 | NA |
| | P13 | (1. Not water) OR (0. No data) |
| | P14 | 0. max_extent |
| | P15 | Not($\geq 1$) |

| | | |
|---|---|---|
| **C27** | P1 | 12. Croplands: at least 60% of area is cultivated cropland. |
| | P2 | 12. Croplands: at least 60% of area is cultivated cropland. |
| | P3 | 3. Broadleaf Croplands: bominated by herbaceous annuals (<2m) that are cultivated with broadleaf crops. |
| | P4 | 5. Annual Broadleaf Vegetation: dominated by herbaceous annuals (<2m). At least 60% cultivated broadleaf crops. |
| | P5 | 8. Broadleaf Croplands: dominated by herbaceous annuals (<2m). At least 60% cultivated broadleaf crops. |
| | P6 | 40. Cultivated and managed vegetation / agriculture. Lands covered with temporary crops followed by harvest and a bare soil period (e.g., single and multiple cropping systems). Note that perennial woody crops will be classified as the appropriate forest or shrub land cover type. |
| | P7 | NA |
| | P8 | 11. Post-flooding or irrigated croplands |
| | P9 | (1. Croplands: irrigation major) OR (2. Croplands: irrigation minor) |
| | P10 | NA |
| | P11 | NA |
| | P12 | NA |
| | P13 | (1. Not water) OR (0. No data) |
| | P14 | 0. max_extent |
| | P15 | Not($\geq 1$) |
| **C28** | P1 | 12. Croplands: at least 60% of area is cultivated cropland. |
| | P2 | 12. Croplands: at least 60% of area is cultivated cropland. |
| | P3 | 3. Broadleaf Croplands: bominated by herbaceous annuals (<2m) that are cultivated with broadleaf crops. |
| | P4 | 5. Annual Broadleaf Vegetation: dominated by herbaceous annuals (<2m). At least 60% cultivated broadleaf crops. |
| | P5 | 8. Broadleaf Croplands: dominated by herbaceous annuals (<2m). At least 60% cultivated broadleaf crops. |
| | P6 | 40. Cultivated and managed vegetation / agriculture. Lands covered with temporary crops followed by harvest and a bare soil period (e.g., single and multiple cropping systems). Note that perennial woody crops will be classified as the appropriate forest or shrub land cover type. |
| | P7 | NA |
| | P8 | 14. Rainfed croplands |
| | P9 | (3. Croplands: rainfed) OR (4. Croplands: rainfed, minor fragments) OR (5. Croplands: rainfed, very minor fragments) |
| | P10 | NA |
| | P11 | NA |
| | P12 | NA |
| | P13 | (1. Not water) OR (0. No data) |
| | P14 | 0. max_extent |
| | P15 | Not($\geq 1$) |

| | | |
|---|---|---|
| | P1 | 13. Urban and Built-up Lands: at least 30% impervious surface area including building materials, asphalt and vehicles. |
| | P2 | 13. Urban and Built-up Lands: at least 30% impervious surface area including building materials, asphalt and vehicles. |
| | P3 | 10. Urban and Built-up Lands: at least 30% impervious surface area including building materials, asphalt and vehicles. |
| | P4 | 8. Urban and Built-up Lands: at least 30% impervious surface area including building materials, asphalt, and vehicles. |
| | P5 | 9. Urban and Built-up Lands: at least 30% impervious surface area including building materials, asphalt, and vehicles. |
| | P6 | 50. Urban / built up. Land covered by buildings and other man-made structures. |
| | P7 | NA |
| C29 | P8 | 190. Artificial surfaces and associated areas (urban areas >50%) |
| | P9 | NA |
| | P10 | NA |
| | P11 | NA |
| | P12 | NA |
| | P13 | (1. Not water) OR (0. No data) |
| | P14 | 0. max_extent |
| | P15 | NU |