# Peer review of "TimeSpec4LULC: A Global Multispectral Time Series Database for Training LULC Mapping Models with Machine Learning"

_Earth System Science Data, 2021_

## Referee Comment (RC1)

The manuscript presents a dataset useful to train machine learning models for LULC mapping and the methods and results it presents are original and sufficiently well explained. However the manuscript would benefit from a clearer focus on the scope and limitations of this training dataset. For instance, both the title and the abstract do not contain 'training' and do not obviously point to the type of data that are presented. Some of the technical choices should be better justified (e.g. why not using the standard Land Cover Classification System as the basis to derive consensus between different land cover and legends; why not including the LCCS-based types from the MODIS land cover; some of the LULC categories are not defined, e.g. broadleaf cropland). The manuscript should more openly discuss the limitations and weaknesses of the dataset including on the methods applied for data validation and on the potential applications of the dataset – for instance it is not immediately obvious if the dataset could support machine learning methods to detect LULC changes with complete transition matrixes. Particularly for the classes with lowest purity – defined as the combined consensus spatially and temporally across products, if would be useful to add to the discussion some insights on what classes cause confusion and reduced consensus. I believe this (ancillary information) may be included as part of classification efforts. This information could be enriched by I recommend the publication of the manuscript after major revision – see my specific comments and suggestions below.

**Title and abstract**: I suggest simplifying the title and adding to the abstract more explicit reference to the actual nature and scope of the dataset. Lines 2-3: It is not entirely true that deep learning networks are unexplored for global mapping efforts (e.g. GHSL built-up areas). In this context, I believe it is more pertinent to focus on the needs for good quality training datasets in all machine learning methods. Incidentally, are there specific reason for not using the JRC GHSL- built up areas in the analysis.

**Introduction**: Line 21 — I suggest including here the definitions of land cover and land use, currently in lines 121-124, also adding appropriate references. Line 26 – unclear what are the biophysical properties of the land use categories. Line 28 – land cover is known as an essential climate (climate missing in the text) variable. You might want to use a synonym instead if climate variable is not appropriate. The sentence is however unclear particularly in the use of 'planet boundary'. I suggest rephrasing. In Table 1, GEE is used in the table but was not defined earlier. In Table 1, JRC Yearly History is more correctly JRC Yearly Water History. More in general, these more technical discussions (detailed reasons for discrepancies between global land cover data; Table 1 and Table 2) should be better placed in Methods (e.g. section 2.1.1). This section should instead provide the general context and present the main objectives of the manuscript and the type of data that is presented.

**Methods:** Figure 1 should be discussed before it is presented.

It is not immediately clear what are the concepts that guide the hierarchical system for the presentation of the LULC classes. The approach seems following the FAO Land Cover Classification System (LCCS) hierarchical approach, but LCCS is never explicitly referred to. This is important because LCCS represents the standard to harmonize land cover legends and it is used in several of the products used in the analysis (e.g. Copernicus land cover; GlobCover) so it appears strange it was not applied in the harmonization of the legends. Also, MODIS LC contains three LCCS-based types (land cover; land use and hydrology) which represent the reference land cover types for this land cover product as defined by its data producers (Sulla-Menashe et al. 2019). It seems strange that they were not included in the analysis.

The LULC categories are not clearly defined. It is especially confusing the separation of cropland classes into cereals; broadleaf and flooded. It is unclear what broadleaf cropland contains (is this permanent/woody crops?). None of the global products used in the dataset contains information on crop type, so it is not immediately clear how cereals were identified. Flooded cropland is not conceptually at the same level as cereals and broadleaf. Likewise, none of the products included the category Mangroves and there is no

indication on the distinction between Swamps and Marshlands. Thresholds for separating open/close coverage are not defined.

Line 134 (caption Table 3): I suggest changing "The numbers from 0 to 220 correspond to the class label in GEE" to "The numbers from 0 to 220 correspond to class values in the original LULC products". A supplementary table describing the characteristics (class values; type of legend; main scope) of the 15 datasets would facilitate the understanding of the rules presented in Table 3. For each dataset, please report the appropriate link in the Data Catalog of the Earth Engine and proper citation when applicable. in the Some of the products contain both discrete and continuous categorization (e.g. Copernicus land cover) with the proportion of land cover classes in the pixel. This might be worth mentioning.

Line 135 – For products that contain only one image (P1 to P7). It is the other way round based on numbering of products in Table 1 and later in Figure 2. Line 142 – Croplands may be hardly defined as a land cover class with high temporal stability. In general, the choice of the best operator for temporal combination could be validated with some sensitivity analysis. Some thoughts should be given in the discussion as to the possible consequences for data quality and applications in the choice of the operator. Line 151 – change 'where applied' to were applied. Lines 151-152 – it is unclear what was done for these 5 classes and what are these classes. Section 2.1.4 and line 156 – resampling seems more accurate terms for this type of spatial operation. I suggest presenting firstly the global results with all the classes (Figure 3) and then the example. The discussion on the method applied to define the purity of the pixel could be improved and better clarified. Currently, the discussion in Lines 160-161 suggests a different meaning of purity than elsewhere in the manuscript. In here, the text refers more to the thematic classification and seems hinting to typical concepts in land cover mapping of pure *vs* mosaic land cover classes whereas elsewhere it explicitly defines purity as the spatial and temporal agreement between the various datasets.

In Figure 3, please use the full name of the classes or refer to Table 6 as you have done elsewhere. Also, it would be useful to add the labels used in the text (C1 to C28), for instance C1 – Barren lands.

Section 2.2.1 – Change 'MODIS sensor is known by..' to 'MODIS sensor has high temporal coverage, ensured by Terra and Aqua satellites revisit frequencies, and also spectral and spatial features that are highly suitable for LULC mapping and change detection..' Also, please provide a supplementary table describing the 7 bands (wavelengths). Line 184 – please explain better the reasons for not using the water flag for Permanent Snow and Cropland. Line 186 – change 'missing-value gaps' to 'missing values'. Line 188 – why the maximum? Is this the highest spectral value for each band? Are there implications associated with this choice?

Line 202 – add link to GAUL dataset in GEE (see also comment above). Line 206-207 – the manuscript does not really explain why the Global modification index was included. Considering how this was produced, there is high risk of multi-collinearity and it's not immediately clear what advantages it brings. It is mentioned later that is gives proves of the good quality of the definition of Built-up areas but this is rather vague. Line 212 – Link in footnote 2 is not working. Footnotes should be better avoided.

Line 231 – I suggest changing 'And, the last 223 columns contain the 223 monthly observations of the time series for one spectral band' TO ', The last 223 columns contain for each point the time series with the 223 monthly surface reflectance. All these values are reported separately for each of the seven spectral bands.

Line 240 – I wonder if 100 pixels are enough to assess the quality of annotation in classes that are less represented. What did guide the choice of this number? It is likely the technical feasibility and availability of resources. Please explain. What are the implications in terms of conclusions on the quality of the dataset?

Lines 241-242 – the applied thresholds may not be appropriate for some land use categories. For instance, in those pixels that contain large proportions of fallow land together with cultivated fields, these thresholds may fail to capture the dominant land use of the pixel.

Line 244 – Please provide reference to the F1 metrics. Please indicate what are the advantages and drawbacks of F1 and if alternative methods exist.

**Results and Discussion** – as discussed above, the limitations of the dataset for global LULC mapping and implications of the various technical choices for potential applications should be more clearly reported in a separate section. For each of the classes with lowest purity – defined in the manuscript as the combined consensus spatially and temporally across products, it would be useful to add to the discussion some insights on what classes exactly cause confusion and reduced consensus. I believe this (ancillary information) may be included in the metadata and support potential classification efforts.

Figure 7 – what variables affect the density of the time series? Considering removing this figure or explain better its usefulness.

Line 246 – report on F1 metrics seems differing to what is reported in the conclusions. Here you wrote 'As it can be noticed, as we go up from level L0 to L5 the obtained dataset accuracy increases from 87% to 96% due mainly to the forests classification'. In lines 298-299 you wrote instead 'The overall accuracy (F1 value) of the annotation varied from 96% at the coarser classification level to 87% at the finest level.'

Table 8 – I am not clear how to interpret the standard deviations that are reported for purity. Please explain.

---

## Author Response (AR1)

**Response letter** for
Manuscript with Ref: essd-2021-253 entitled
"TimeSpec4LULC: A Global Multispectral Time Series Database for Training
LULC Mapping Models with Machine Learning"

We would like to thank reviewers 1 and 2 for the valuable comments. We took all the suggestions into account and highlighted all changes in blue color in the revised manuscript. Below we provide a list of the main changes and then we explain in detail how we addressed each one of the reviewers' comments.

Herein, a list of the main changes in the revised manuscript:

1. In Section 5 (Advantages, Limitations, and Potential Applications of the dataset), we now include the advantages, limitations and potential applications of the dataset.
2. To justify the choice of the agreement threshold for each class as well as to highlight the classes that reached lower consensus (those where we had to reduce the agreement threshold to find enough number of representative pixels for it), we now include in Table 7 a sensitivity analysis of the number of eventually selected pixels resulting from the application of different thresholds of agreement for each class.
3. To better explain 1) the temporal (inter-annual) combination within each product, 2) the spatial combination across products, 3) the selection of the agreement threshold, and 4) the selection of 1000 evenly distributed pixels, we now include four boxes with the four corresponding algorithms (rulesets written as pseudocode in Algorithms 1, 2, 3 and 4, respectively).
4. In this new version, instead of using the Maximum reflectance for each month, we now use the Mean operator in the monthly aggregation. In addition, we have added the period from 05-03-2000 to 04-07-2002 using only the MODIS-Terra time series (since Aqua was launched 2 years later than Terra), and the period from 04-07-2002 to 01-01-2022 using Terra+Aqua time series (we added the full year 2021). Hence, the acquisition time of the dataset is now covering 22 years. The old version has been deleted and substituted by this new version of the dataset.
5. To help the users explore the dataset, we now provide two versions of the data: the original dataset, and a balanced subset of the original dataset based on 1000 samples from each class, evenly distributed over the globe, selected by Algorithm 4.
6. To facilitate the understanding of the definition and nomenclature of the land-cover classes, we now provide in the Appendix the equivalence between the dataset nomenclature and the FAO's Land Cover Classification System (LCCS) in Table A1, and a detailed definition of each LULC class based on the 15 LULC products in Table A2.

**Reviewer #1**

The manuscript presents a dataset useful to train machine learning models for LULC mapping and the methods and results it presents are original and sufficiently well explained. However, the manuscript would benefit from a clearer focus on the scope and limitations of this training dataset. For instance, both the title and the abstract do not contain 'training' and do not obviously point to the type of data that are presented. Some of the technical choices should be better justified (e.g. why not using the standard Land Cover Classification System as the basis to derive consensus between different land cover and legends ; why not including the LCCS-based types from the MODIS land cover ; some of the LULC categories are not defined, e.g. broadleaf cropland . The manuscript should more openly discuss the limitations and weaknesses of the dataset including on the methods applied for data validation and on the potential applications of the dataset – for instance it is not immediately obvious if the dataset could support machine learning methods to detect LULC changes with complete transition matrixes. Particularly for the classes with lowest purity – defined as the combined consensus spatially and temporally across products, it would be useful to add to the discussion some insights on what classes cause confusion and reduced consensus. I believe this (ancillary information) may be included as part of classification efforts. This information could be enriched by I recommend the publication of the manuscript after major revision – see my specific comments and suggestions below.

**Authors response to general comments**

- *The Title and Abstract were modified to highlight the scope of this dataset (see responses to specific comment C1 below).*
- *Section 5 was included to highlight the limitations of this dataset, such as that the dataset does not provide transition matrices (see responses to specific comment C1 in Result and Discussion), and its potential applications.*
- *The hierarchical structure of the LULC classes was created based on the FAO's LCCS, see Table A1 in Appendix with a dictionary of equivalences in naming and definitions of each class (see responses to specific comment C2 in Methods). Several LULC classifications were used, including the five LULC classifications used by MODIS (see Table 2).*
- *The definitions of each LULC class with respect to each product were provided in Table A2 in Appendix (see responses to specific comments C2, C4, C5, and C7 in Methods).*
- *Insights about the classes that had reduced the spatial and temporal consensus were provided in Section 4 and Table 7 (see responses to specific comment C2 in Result and Discussion).*

**Authors response to specific comments**

**Title and abstract**:

C1. I suggest simplifying the title and adding to the abstract more explicit reference to the actual nature and scope of the dataset.

**Authors response**

We have changed the title, in the revised manuscript, from ''*TimeSpec4LULC: A Global Deep Learning-driven Dataset of MODIS Terra-Aqua Multi-Spectral Time Series for LULC Mapping and Change Detection*'' to ''*TimeSpec4LULC: a global multispectral time series database for training LULC mapping models with machine learning*''.

In addition, we have explicitly mentioned, in the abstract in the revised manuscript, that the dataset is targeted towards training machine learning models. Now, both the title and the abstract, in the revised manuscript, provide detailed information about the nature and scope of the dataset.

C2. Lines 2-3: It is not entirely true that deep learning networks are unexplored for global mapping efforts (e.g. GHSL built-up areas). In this context, I believe it is more pertinent to focus on the needs for good quality training datasets in all machine learning methods.

**Authors response**

Thank you for this clarification. Now, we mention in the abstract of the revised manuscript that the best way to create accurate LULC maps is by building good quality state-of-the-art machine learning models, and that high-quality training datasets are required to build such models. We also highlight in the revised manuscript that our dataset provides a good-quality training time series thanks to the spatial and temporal agreement over 15 global LULC products (see abstract).

C3. Incidentally, are there specific reasons for not using the JRC GHSL- built up areas in the analysis.

**Authors response**

We did not use this product because it was not available in GEE when we carried out our study. We will include it in future versions of the dataset since it would involve reprocessing everything from the beginning.

**Introduction**:
1. Line 21 — I suggest including here the definitions of land cover and land use, currently in lines 121-124, also adding appropriate references.

**Authors response**

The land cover and the land use definitions have been moved to the first paragraph of the introduction in the revised manuscript.

2. Line 26 – unclear what are the biophysical properties of the land use categories.

**Authors response**

We explained that the LULC change alters the climate through two mechanisms: the biophysical (BPH) and the biogeochemical (BGC) feedbacks. We provided the difference between the two mechanisms using the example of the forest conversion to croplands. We made this idea clearer in the second paragraph, in section 2, in the revised manuscript.*"*

3. Line 28 – land cover is known as an essential climate (climate missing in the text) variable. You might want to use a synonym instead if climate variable is not appropriate. The sentence is however unclear particularly in the use of 'planet boundary'. I suggest rephrasing.

**Authors response**

We have rewritten this sentence in the 2nd paragraph, in section 1, in the revised manuscript, and have incorporated citations for the definitions of essential climate variables and essential biodiversity variables. We also include a citation for the Planetary boundaries concept to find a safe operating space for humanity.*"*

4. In Table 1, GEE is used in the table but was not defined earlier.

**Authors response**

Google Earth Engine and its acronym are now defined in the abstract, the first time it is used in the main text, and in all figure captions and table headings, in the revised manuscript.

5. In Table 1, JRC Yearly History is more correctly JRC Yearly Water History.

**Authors response**

Thank you! The name of this product was corrected in Table 2, in the revised manuscript.

6. More in general, these more technical discussions (detailed reasons for discrepancies between global land cover data: Table 1 and Table 2) should be better placed in Methods (e.g. section 2.1.1). This section should instead provide the general context and present the main objectives of the manuscript and the type of data that is presented.

**Authors response**

As suggested by the reviewer, an explanation about LULC products limitations and discrepancies along with Table 1 (currently named Table 2) were moved to section 2.1 (*Finding spatial-temporal agreement across 15 global LULC products*), in the revised manuscript. The potential of deep learning for LULC mapping, the review of available LULC datasets to train deep learning models, and Table 2 (currently named Table 1) were maintained in section 1 (*Introduction*), in the revised manuscript. This is to provide the reader with the motivation behind our study and the limitations of the existing DL datasets.

**Methods:**

1. Figure 1 should be discussed before it is presented.

**Authors response**

In the revised manuscript, Figure 1 is now discussed then presented in section 2.1.2.

2. It is not immediately clear what are the concepts that guide the hierarchical system for the presentation of the LULC classes. The approach seems following the FAO Land Cover Classification System (LCCS) hierarchical approach, but LCCS is never explicitly referred to. This is important because LCCS represents the standard to harmonize land cover legends and it is used in several of the products used in the analysis (e.g. Copernicus land cover; GlobCover) so it appears strange it was not applied in the harmonization of the legends. Also, MODIS LC contains three LCCS-based types (land cover; land use and hydrology) which represent the reference land cover types for this land cover product as defined by its data producers (Sulla-Menashe et al. 2019). It seems strange that they were not included in the analysis.

**Authors response**

Thank you very much for pointing this out. Wrongly, we implicitly assumed that it was clear that we were using FAO's Land Cover Classification System since we were using LC products that use such a system. The hierarchical structure of our LULC classes and the correspondence between our nomenclature and FAO's system is now explicitly explained in paragraph 2, section 2.1.2

(2.1.2 Standardization and Harmonization of LULC legends). We also included a new Table A1 in the Appendix with the correspondence between our nomenclature and FAO's LCCS system. Further, the five LULC classification systems from MODIS used to build the nomenclature are referenced in Table 2, in the revised manuscript. Rather than create another classification system, we wanted to find the lowest common denominator set that was classification-free yet allowed for approximation and cross-walking
of multiple classifications at their fundamental levels using FAO's LCCS as a reference.

3.      The LULC categories are not clearly defined. It is especially confusing the separation of cropland classes into cereals; broadleaf and flooded. It is unclear what broadleaf cropland contains (is this permanent/woody crops?). None of the global products used in the dataset contains information on crop type, so it is not immediately clear how cereals were identified. Flooded cropland is not conceptually at the same level as cereals and broadleaf.

**Authors response**

Thank you for this suggestion of including a clearer and explicit definition for each class. This is now solved by providing the definitions of each class in Table A2, in Appendix, in the revised manuscript. We agree with the reviewer in that some of the resulting subclasses are not conceptually at the same level, though this is now solved by providing the equivalence to FAO's LCCS hierarchy, so the user can ungroup or regroup our categories at his/her convenience following FAO's LCCS or any other classification scheme. The reason behind our resulting class nomenclature is that we followed a functional approach to identify our classes based on the information that was available in the original LULC products, given that the consensus across products could retriev a representative number of pixels. Sometimes all LULC products used approximately the same broad nomenclature and definition for a particular LULC class, but some other times some products had a more detailed or specific legend than others. For instance, products P8 (GLOBCOVER) and P9 (GFSAD) differentiate between rainfed and irrigated croplands.

Products P3, P4, and P5 (MCD12Q1 types 3, 4, and5, respectively) provide information to differentiate between broadleaf and cereal croplands. Hence, to annotate broadleaf croplands in our dataset, we used the following MCD12Q1 categories: *'Broadleaf croplands'* in P3 and P5, and as *'Annual Broadleaf Vegetation'* in P4. Cereal croplands were defined as '*Grasslands dominated by herbaceous annuals (<2m) including cereal croplands*' in P3, *'Annual Grass Vegetation dominated by herbaceous annuals (<2m) including cereal croplands'* in P4, and *'Cereal Croplands dominated by herbaceous annuals (<2m), at least 60% cultivated cereal crops'* in P5.

4.      Likewise, none of the products included the category Mangroves and there is no indication on the distinction between Swamps and Marshlands.

**Authors response**

Likewise, this is now solved by providing the definitions of each class in Table A2, in Appendix, in the revised manuscript. To differentiate between marshlands, swamps and mangroves, we used the percentage of tree canopy cover in P7 and P11, and tree heights in P12, since marshlands do not have trees while mangroves and swamps do have trees. To differentiate between swamps and mangroves, we used product P8, which differentiates forests on saline waters from forests on freshwaters. That is, to define swamps, we used ((180. Closed to open (>15%) vegetation

(grassland, shrubland, woody vegetation) on regularly flooded or waterlogged soil - fresh, brackish or saline water) OR (160. Closed (>40%) broadleaved forest regularly flooded - Fresh water)). To define mangroves, we used (170. Closed (>40%) broadleaved semi-deciduous and/or evergreen forest regularly flooded - saline water).

5.      Thresholds for separating open/close coverage are not defined.
**Authors response**
We chose these adjectives from the existing products and maintained the same thresholds of shrub or tree coverage. This information was made clearer by providing the definitions of each class in Table A2, in Appendix, in the revised manuscript. For instance, in the case of shrublands, the definitions of closed/open shrublands are provided by P1, P2, and P8. That is, closed shrublands were also defined as *'closed shrublands'* in P1 and P2, and *'Closed to open (>15%) shrubland (<5m)'* in P8. Likewise, open shrublands were defined as *'open shrublands'* in P1 and P2, and *'Sparse (>15%) vegetation (woody vegetation, shrubs, grassland)'* in P8.

6.      Line 134 (caption Table 3): I suggest changing "The numbers from 0 to 220 correspond to the class label in GEE" to "The numbers from 0 to 220 correspond to class values in the original LULC products".
**Authors response**
We thank the reviewer for this observation. Actually, these values correspond to the class Ids (Identification numbers). The class Id was used instead of class value in Table 4 and all over the revised manuscript.

7.      A supplementary table describing the characteristics (class values; type of legend; main scope) of the 15 datasets would facilitate the understanding of the rules presented in Table 3.
**Authors response**
Thanks again. The suggested information was provided in Table A2, in Appendix, in the revised manuscript.

8.      For each dataset, please report the appropriate link in the Data Catalog of the Earth Engine and proper citation when applicable.
**Authors response**
The reference and link (via hyperlink) of each product were provided in Table 2 in the revised manuscript.

9.      In some of the products contain both discrete and continuous categorization (e.g. Copernicus land cover) with the proportion of land cover classes in the pixel. This might be worth mentioning.
**Authors response**
The discrete/continuous categorization of LULC products is now explained in section 2.1.2 in the revised manuscript, as follows:
*"Some of the products provide discrete categorizations of LULC classes in each pixel (P1-P5, P8-P10, P13, and P14), while other products provide continuous categorizations represented by a*

*class proportion in each pixel (P11, P12, and P15), or even both continuous and discrete categorizations of LULC (P6 and P7) (Table 4). To define the class of each pixel within these different categorization mechanisms, we either specify a unique value (e.g., select the value 16 to access barren lands in P1) or use a range of values (e.g., Tree Canopy Cover less than 10 (TCC <10) to access barren lands in P6)"*

10. Line 135 – For products that contain only one image (P1 to P7). It is the other way round based on numbering of products in Table 1 and later in Figure 2.

**Authors response**

This mistake was corrected in Table 2 in the revised manuscript. As explained in the previous comment, the discrete/continuous categorization of LULC products is now explained in section 2.1.2 in the revised manuscript.

11. Line 142 – Croplands may be hardly defined as a land cover class with high temporal stability. In general, the choice of the best operator for temporal combination could be validated with some sensitivity analysis.

**Authors response**

We agree that some LULCs have inherent high inter-annual variability, like croplands. Our general objective was to collect from each class a representative number of pixels (at least 1000) that satisfy the temporal stability constraint of a specific class type. Given this objective, we performed the temporal combination using two different operators: (1) The AND operator, which represents strict temporal stability constraint, to ensure getting pixels with stable class type over time but more likely small number of pixels. (2) When the AND operator resulted in less than 1000 pixels, we used the MEAN operator, which represents a softer temporal stability constraint, to provide a large enough number of pixels, however, with less temporal stability. The usage of these two operators is governed by the following algorithm (see Algorithm 1). Even though croplands are characterized by high variability over time, when we applied the AND operator we got a large number of pixels (see Table 7). Thus, we preferred to provide this number of pixels characterizing a more stable time series pattern for each class, instead of providing too many pixels (using the MEAN operator) with noisy patterns due to class instability over time. That means that at the resolution of the input datasets, croplands in our dataset were classified as such throughout all years.

The discussion about the usage of temporal combination operators was provided in paragraph 2, in section 2.1.3, in the revised manuscript. Two algorithms describing clearly the process of the temporal and spatial combination were also provided in Algorithm 1 and Algorithm 2, in the revised manuscript.

12. Some thoughts should be given in the discussion as to the possible consequences for data quality and applications in the choice of the operator.

**Authors response**

As explained in the previous point, the discussions about the applications and the consequences of using the operators/rules in the temporal/spatial combination along with two algorithms describing these two processes were provided in section 2.1.3, in the revised manuscript.

13.    Line 151 – change 'where applied' to were applied.

**Authors response**

This mistake was corrected in the last paragraph, in section 2.1.3, in the revised manuscript.

14.    Lines 151-152 – it is unclear what was done for these 5 classes and what are these classes.

**Authors response**

The reason behind using 6 different rules was explicitly explained in Algorithm 2, in the revised manuscript. In addition, the rule type used for each of the five classes was described in the last paragraph, in section 2.1.3, in the revised manuscript.

15.    Section 2.1.4 and line 156 – resampling seems more accurate terms for this type of spatial operation. I suggest presenting firstly the global results with all the classes (Figure 3) and then the example.

**Authors response**

As suggested by the reviewer, Figure 2 (previously named Figure 3) of the global LULC mask was presented in section 2.1.4, in the revised manuscript, then the example was presented in Figure 3 (previously named Figure 2), in the revised manuscript. We also include the term resampling.

16.    The discussion on the method applied to define the purity of the pixel could be improved and better clarified.

**Authors response**

The discussion about the pixel agreement threshold (previously defined as purity) was improved and clarified in section 2.1.4, in the revised manuscript.

17.    Currently, the discussion in Lines 160-161 suggests a different meaning of purity than elsewhere in the manuscript. In here, the text refers more to the thematic classification and seems hinting to typical concepts in land cover mapping of pure *vs* mosaic land cover classes whereas elsewhere it explicitly defines purity as the spatial and temporal agreement between the various datasets.

**Authors response**

The meaning of the agreement (previously named as spatial purity) was corrected in section 2.1.4, in the revised manuscript. To remove confusion, both terms referring to the temporal purity and spatial purity were changed, through all the revised manuscript, as follows:

- Temporal purity: was renamed as the temporal stability of one pixel over time.
- Spatial purity: was renamed as the spatial agreement over the 15 LULC products on a specific pixel.

18. In Figure 3, please use the full name of the classes or refer to Table 6 as you have done elsewhere. Also, it would be useful to add the labels used in the text (C1 to C28), for instance C1 – Barren lands.

**Authors response**

The labels (C1 to C29) were included beside the short names, in Figure 2, in the revised manuscript. The long names were not used because of the space constraints in the map's legend. Then, Table 3 was cited in the caption of Figure 2 to help the reader track the full names.

19. Section 2.2.1 – Change 'MODIS sensor is known by..' to 'MODIS sensor has high temporal coverage, ensured by Terra and Aqua satellites revisit frequencies, and also spectral and spatial features that are highly suitable for LULC mapping and change detection..'

**Authors response**

This sentence was corrected in the first paragraph, in section 2.2.1, in the revised manuscript.

20. Also, please provide a supplementary table describing the 7 bands (wavelengths).

**Authors response**

A description of these bands is now provided in Table 6, in the revised manuscript.

21. Line 184 – please explain better the reasons for not using the water flag for Permanent Snow and Cropland.

**Authors response**

The reason for not using the water flag for Permanent Snow and Croplands Flooded with Seasonal Water was explained, in the last paragraph of section 2.2.1, in the revised manuscript, as follows: "*The water flag (bits 3-5) was used to mask out water pixels in all terrestrial systems, but not in the terrestrial systems of Permanent Snow, and in Croplands Flooded with Seasonal Water to avoid unrealistic data loss*"

22. Line 186 – change 'missing-value gaps' to 'missing values'.

**Authors response**

This expression was changed in section 2.2.2, in the revised manuscript.

Line 188 – why the maximum? Is this the highest spectral value for each band? Are there implications associated with this choice?

**Authors response**

Thank you for this critical comment. Initially, we used the Max function on the single-NDVI band and later we decided to use multispectral bands instead on NDVI and forgot to change Max to Mean. Now, we re-exported the time series data using the Mean value and updated the corresponding description in section 2.2.2, in the revised manuscript.

23.     Line 202 – add link to GAUL dataset in GEE (see also comment above).
**Authors response**
We included all the URL-links as references in the References section in the revised manuscript. References to the links of both GAUL products ADM0 and ADM1 were added in section 2.2.4, in the revised manuscript.

24.     Line 206-207 – the manuscript does not really explain why the Global modification index was included. Considering how this was produced, there is high risk of multi-collinearity and it's not immediately clear what advantages it brings. It is mentioned later that it gives proves of the good quality of the definition of Built-up areas but this is rather vague.
**Authors response**
The way this index was produced and its relevance as a metadata were provided, in section 2.2.4, in the revised manuscript, as follows:
"*To provide the user with extra metadata that could be used to filter time series according to different levels of human intervention on each pixel, the GHM index was included. The GHM index was derived from the Global Human Modification dataset (CSP gHM) (Kennedy et al.,2019) available in GEE, which provides a cumulative measure of human modification of terrestrial lands. Then, it was projected to MODIS resolution using the spatial mean reducer to generate the average GHM index.*"

25.     Line 212 – Link in footnote 2 is not working. Footnotes should be better avoided.
**Authors response**
All the footnotes were deleted, in the revised manuscript, and were substituted by references (e.g., see section 3). We included all the URL-links as references in the References section in the revised manuscript.

26.     Line 231 – I suggest changing 'And, the last 223 columns contain the 223 monthly observations of the time series for one spectral band' TO ', The last 223 columns contain for each point the time series with the 223 monthly surface reflectance. All these values are reported separately for each of the seven spectral bands.
**Authors response**
Section 3.1 was re-written, in the revised manuscript since the data structure was changed from csv to json as well as the time series length was expanded from 223 to 262.

27.     Line 240 – I wonder if 100 pixels are enough to assess the quality of annotation in classes that are less represented. What did guide the choice of this number? It is likely the technical feasibility and availability of resources. Please explain.
**Authors response**
We agree with the reviewer that the number of 100 validated pixels per class is relatively small, particularly regarding some classes containing a high number of pixels. As the reader suggests, the choice of this number was due to the challenging technical feasibility of the validation process and

the lack of control resources. However, the pixels of each class were randomly selected following the maximum distance criteria which makes them spatially representative of each class. This limitation was discussed in section 5, in the revised manuscript.

28.     What are the implications in terms of conclusions on the quality of the dataset?

**Authors response**

The quality of the dataset was discussed, in the first paragraph, in section 5, in the revised manuscript, as follows:

*"The produced dataset is of high quality both in terms of the annotation and the generation of spectral reflectance. On the one hand, our dataset was annotated using the process of spatial-temporal combination of 15 global LULC products available in GEE. On the other hand, the time series of spectral reflectance were generated with less noise thanks to (1) the application of the quality assessment filters (MODLAND QA and State QA) in both MODIS products (MOD09A1 and MYD09A1), (2) the temporal aggregation from 8-day to monthly data, and (3) the Terra+Aqua merging process. In addition, the annotation accuracy was assessed in two ways. First, thanks to the spatial-temporal agreement across the global GEE products, the level of consensus offered a cross-validation across independent products. Second, using a geographically representative sample of 2900 pixels (100 pixels per class selected by Algorithm 4) manually inspected by experts (visually photo-interpreted) using very high resolution imagery from both Google Earth and Bing Maps. Then, jointly agreed on which class each pixel corresponded to (agreement across interpreters according to Muchoney et al. (1999)). Thus, the high quality of this dataset will certainly ensure the building of highly accurate machine learning models because building good quality machine learning models is possible only when trained on good quality data (García-Gil et al., 2019)."*

29.     Lines 241-242 – the applied thresholds may not be appropriate for some land use categories. For instance, in those pixels that contain large proportions of fallow land together with cultivated fields, these thresholds may fail to capture the dominant land use of the pixel.

**Authors response**

We agree with the reviewer. In a cultivated landscape, some plots may be in rotative fallow while other plots are being cultivated. Even though, the main signal from this class would come from the cultivated land (which is the main land-use of the pixel). In any case, this limitation was discussed in the last paragraph in section 4, in the revised manuscript.

30.     Line 244 – Please provide reference to the F1 metrics. Please indicate what are the advantages and drawbacks of F1 and if alternative methods exist.

**Authors response**

The definition of the F1 score, its limitations and advantages were provided in section 3.2, in the revised manuscript.

**Results and Discussion**:

1. As discussed above, the limitations of the dataset for global LULC mapping and implications of the various technical choices for potential applications should be more clearly reported in a separate section.

**Authors response**

A new section about the advantages, the limitations, and the potential applications of the dataset was included in section 5, in the revised manuscript.

2. For each of the classes with lowest purity – defined in the manuscript as the combined consensus spatially and temporally across products, it would be useful to add to the discussion some insights on what classes exactly cause confusion and reduced consensus. I believe this (ancillary information) may be included in the metadata and support potential classification efforts.

**Authors response**

A sensitivity analysis of the number of pixels with respect to different thresholds of agreement, along with the final number of collected pixels at each selected threshold for the 29 LULC classes is now provided in Table 7, in the revised manuscript.

The classes that most importantly reduced the consensus accuracy (the three Wetlands classes, all the Forests classes except class C8 and C14, and the Closed Shrublands) were highlighted and discussed in the first paragraph in section 4, and in section 5, in the revised manuscript.

The pixel agreement percentage was also provided as a metadata so that the user can control the desired threshold and take it into consideration to evaluate the accuracy of the models. The user can now further increase (decrease) the threshold agreement and subsequently reduce (augment) the number of selected pixels at his/her convenience.

3. Figure 7 – what variables affect the density of the time series? Considering removing this figure or explain better its usefulness.

**Authors response**

This figure was removed from the revised manuscript because we assumed that the new Figure 2 presents enough information about the distribution of the data. Figure 2 currently shows in which place of the world the 29 LULC classes are more stable in time, and the 15 LULC products are more compliant, since the number of the collected pixels in each class is affected by the temporal stability of the 29 LULC classes and the spatial agreement over the 15 LULC products.

4. Line 246 – report on F1 metrics seems differing to what is reported in the conclusions. Here you wrote 'As it can be noticed, as we go up from level L0 to L5 the obtained dataset accuracy increases from 87% to 96% due mainly to the forests classification'. In lines 298-299 you wrote instead 'The overall accuracy (F1 value) of the annotation varied from 96% at the coarser classification level to 87% at the finest level.'

**Authors response**

Thank you! This mistake was corrected in section 3.2, in the revised manuscript.

5.     Table 8 – I am not clear how to interpret the standard deviations that are reported for purity. Please explain.

**Authors response**

Certainly, reporting the standard deviations was confusing and not informative so they were removed from Table 10, in the revised manuscript.

**Reviewer #2**

**General comments**

The manuscript is generally well written and organized. I recommend it for publication if the following are addressed. This does not require further processing and/or analysis, just clarification.

1. There are no reference to MODIS LC product at all nor of the ATBD: Strahler, A., D. Muchoney, J. Borak, G. Feng, M. Friedl, S. Gopal, J. Hodges, E. Lambin, D. McIver, A. Moody, C. Schaaf, and C. Woodcock. 1999. MODIS Land Cover Product Algorithm Theoretical Basis Document (ATBD) Version 5.0. Boston: Boston University, 89 pp.

**Authors response**

The link and the reference of all the used LULC products (including the reference above) are now provided in Table 2, in the revised manuscript.

2. Per validation / accuracy assessment: Expert opinion is extremely problematic. There is no mention as to cross-validation, agreement of interpreters etc. Is this just a call as to what class it might be? (Muchoney, D.M., A. Strahler, J. Hodges and J. Locastro. 1999. The IGBP DISCover Confidence Sites and the System for Terrestrial Ecosystem Parameterization: Tools for Validating Global Land Cover Data. Photogrammetric Engineering and Remote Sensing 65 (9): 1061-1067).

**Authors response**

We completely agree on the challenge of assessing or validating accuracy of LULC annotations. We followed two ways to assess accuracy in the LULC annotations of our dataset. First, our dataset uses spatial and temporal agreement across 15 LULC products (cross-validation) as the main pathway to gain accuracy in LULC annotation (i.e., accuracy as consensus among the 15 LULC products). Additionally, we use our own expert interpretation to report on the eventual accuracy reached by the former procedure (spatio-temporal agreement across products). For this, we selected from each of the 29 classes a set of 100 random samples representatively distributed around the world by maximizing the distance among sampled pixels. Then, these 2900 images were manually validated (visual photo-interpretation) by two experts who jointly agreed on which class each pixel corresponded to (agreement of interpreters according to Muchoney et al. 1999) by using very high resolution images from both Google Earth and Bing Maps. Since such a validation process was technically challenging and very time-consuming, this task was performed only on 100 images per class.

The explanation was included in the manuscript in Section 5 as follows:

*"The produced dataset is of high quality both in terms of the annotation and the generation of spectral reflectance. On the one hand, our dataset was annotated using the process of spatial-temporal combination of 15 global LULC products available inGEE. On the other hand, the time series of spectral reflectance were generated with less noise thanks to (1) the application of the quality assessment filters (MODLAND QA and State QA) in both MODIS products (MOD09A1 and MYD09A1), (2) the temporal aggregation from 8-day to monthly data, and (3) the Terra+Aqua merging process.In addition, the annotation accuracy was assessed in two ways. First, thanks to the spatial-temporal agreement across the global GEE products, the level of consensus offered a cross-validation across independent products. Second, using a geographically representative sample of 2900 pixels (100 pixels per class selected by Algorithm 4) manually inspected by experts (visually photo-interpreted) using very high resolution imagery from both Google Earth and Bing*

*Maps. Then, jointly agreed on which class each pixel corresponded to (agreement across interpreters according to Muchoney et al. (1999)). Thusthe high quality of this dataset will certainly ensure the building of highly accurate machine learning models because building good quality machine learning models is possible only when trained on good quality data (García-Gil et al., 2019)."*

3.  Co-registration: There is no documentation of pixel-to-pixel co-registration. Did you look at the PSF?

**Authors response**

Unfortunately, we could not look at the PSF because the co-registration is automatically achieved by GEE. When you operate two different images from two different sources (products) they are already coregistered by Google. When these source images have different scales, the generated image always inherits the resolution of the last image.

We now mention, in the first paragraph of section 2.1.4, in the revised manuscript, that the final mask of each LULC class maintained the spatial resolution of the last aggregated LULC product (P15) at the finest (30m) resolution.

4.  There are no problems reported: Might there not be mention of assumptions, possible errors?

**Authors response**

Thank you for this great suggestion. A new section about the advantages, limitations, and potential applications of the dataset is now included in section 5, in the revised manuscript.

5.  There is no details on the classification algorithms.

**Authors response**

References to each LULC product are now provided in Table 2, in the revised manuscript, so that the reader can have more details about the classification approach used within each LULC product. To facilitate the understanding of the definition and nomenclature of the LULC classes of our dataset, we now provide in the Appendix the equivalence between the dataset nomenclature and the FAO's Land Cover Classification System (LCCS) in Table A1, and a detailed definition of each LULC class based on the 15 LULC products in Table A2. The algorithms that we followed to build each class can be found in Table 4 and Table 5.

6.  Monthly composites: Are not 32-day monthly MODIS composites available?

**Authors response**

There used to be a MODIS 32-day composite data but it was deprecated and it is not included in GEE. Currently, in GEE only MODIS surface reflectance data for the seven bands is available in 8-day composites at 500m and 1km.

**Specific comments**

1. 10 I don't know what "smartly pre-processed" means

**Authors response**

A definition of smartly-preprocessed data, is now provided in section 1, in the revised manuscript, as follows:

"The concept of smart data involves all pre-processing methods that improve value and veracity of data and of associated expert annotations (Luengo et al., 2020) resulting in high quality and accurately annotated datasets. In general, remote sensing datasets contain noise, missing values, and high variability and complexity across space, time, and spectral bands. Applying pre-processing methods, such as gap filling and noise reduction to data, and consensus across multiple sources to annotations, contribute to creating smart remote sensing datasets."

2. 25 Not sure what an "essential planetary boundary" might be

**Authors response**

The sentence that contained this expression was rephrased, in section 1, in the revised manuscript, as follows:

"LULC is an essential climate and biodiversity variable (Bojinski et al., 2014; Pettorelli et al., 2016) to model and assess the status and trends of social-ecological systems from the local to the global scale in the pursuit of a safe operating space for humanity (Steffen et al., 2015)."

3. 45 Global products are not meant for local studies. MODIS was meant to parameterize BGC and GCM.

**Authors response**

We completely agree. Here, we mean that the accuracy of the global products at the local level is low compared to their accuracy at the global level and to the accuracy of local products at local level. We have rephrased this sentence in the first paragraph of Section 2.1.

4. 60 Deep Learning is not defineds.

**Authors response**

A definition of deep learning was included, in paragraph 4, in section 1, in the revised manuscript, as follows:

"Deep Learning (DL), a sub-field of machine learning essentially based on deep artificial neural networks (Zhang et al., 2018c), has ....".

5. 80 CNN and RNN are not described. NNs have been in use for quite some time, at least 20 years

**Authors response**

We agree that this contextual information is needed. We now mention in paragraph 4 of section 1, in the revised manuscript, that CNNs are used to extract spatial patterns, while RNNs are used to extract temporal/sequential patterns. In addition, the sentence mentioning the NNs usage was corrected, in the revised manuscript, as follows:

"Deep Learning (DL), a sub-field of machine learning essentially based on deep artificial neural networks, has shown impressive performance in computer vision and promising ones in remote sensing during the last decades."

6. 90 The purpose overall is to create a dataset that allows for deep learning?
**Authors response**
We clarify this in the abstract and in Section 5. The objective is to create a dataset that can be used to develop and assess LULC products but it is particularly designed for machine learning models, more specifically, for deep learning models. In any case, the dataset allows for global scale analysis or modelling of many LULC classes using long time series data.

7. 125 Cross-walking categorical/nominal variables, i.e. classes is deeply problematic
**Authors response**
The class Id, the class full name, and the class short name were provided in Table 3, in the revised manuscript. In Table 4 and Table 5, we clearly specify how we combined categorical and numerical LULC products to derive each class in our dataset. To facilitate the understanding of such definition and the nomenclature of the land-cover classes, we now provide in the Appendix the equivalence between the dataset nomenclature and the FAO's Land Cover Classification System (LCCS) in Table A1, and a detailed definition of each LULC class based on the 15 LULC products in Table A2.

8. 185 What about areas with persistent cloud cover: Ecuador, Colombia, Peru, DRC, ROC?
**Authors response**
If any pixel with persistent cloud cover was annotated for a particular LULC, it will appear in the metadata with on coordinates, their class labels, and other information. However, if cloud coverage was too persistent, the time-series would be empty due to the application of Modis cloud mask.

9. 200 Again, global products are not necessarily meant to be down-scaled
**Authors response**
The final mask was generated at 30 m resolution because the last product applied has 30m resolution. Thus, it was reprojected to Modis resolution (500 m) in order to extract the time series at Modis resolution.

**Technical corrections Graphics:**
1. While this may just be the pdf rendering, the graphics are small and color rendering dull: Figures: 2, 3, 4 5
**Authors response**
Thank you for letting us know. The colors were changed, and the size and the resolution of the images were increased, in the revised manuscript.